# Liver-directed lentiviral gene therapy corrects hemophilia A mice and achieves normal-range factor VIII activity in non-human primates

Michela Milani [1], Cesare Canepari[1,2], Tongyao Liu[3], Mauro Biffi[1], Fabio Russo[1], Tiziana Plati[1], Rosalia Curto[1], Susannah Patarroyo-White[3], Douglas Drager[3], Ilaria Visigalli[1], Chiara Brombin[4], Paola Albertini[1], Antonia Follenzi [5], Eduard Ayuso[6], Christian Mueller[3], Andrea Annoni [1], Luigi Naldini [1,2,7 ✉] & Alessio Cantore [1,2,7 ✉]

Liver gene therapy with adeno-associated viral (AAV) vectors delivering clotting factor transgenes into hepatocytes has shown multiyear therapeutic benefit in adults with hemophilia. However, the mostly episomal nature of AAV vectors challenges their application to young pediatric patients. We developed lentiviral vectors, which integrate in the host cell genome, that achieve efficient liver gene transfer in mice, dogs and non-human primates, by intravenous delivery. Here we first compare engineered coagulation factor VIII transgenes and show that codon-usage optimization improved expression 10-20-fold in hemophilia A mice and that inclusion of an unstructured XTEN peptide, known to increase the half-life of the payload protein, provided an additional >10-fold increase in overall factor VIII output in mice and non-human primates. Stable nearly life-long normal and above-normal factor VIII activity was achieved in hemophilia A mouse models. Overall, we show long-term factor VIII activity and restoration of hemostasis, by lentiviral gene therapy to hemophilia A mice and normal-range factor VIII activity in non-human primate, paving the way for potential clinical application.

[1] San Raffaele Telethon Institute for Gene Therapy, IRCCS San Raffaele Scientific Institute, Milan, Italy. [2] Vita-Salute San Raffaele University, Milan, Italy. [3] Sanofi, Waltham, MA, USA. [4] University Center for Statistics in the Biomedical Sciences, Vita-Salute San Raffaele University, Milan, Italy. [5] Department of Health Sciences, University of Piemonte Orientale, Novara, Italy. [6] INSERM UMR1089, University of Nantes, CHU de Nantes, 44093 Nantes, France. [7] These authors jointly supervised this work: Luigi Naldini, Alessio Cantore. ✉email: naldini.luigi@hsr.it; cantore.alessio@hsr.it

Hemophilia is an inherited bleeding disorder due to mutations in *F8* or *F9* genes encoding for factor VIII (FVIII) or factor IX (FIX) protein, respectively, which are necessary factors for proper blood coagulation and hemostasis. People with severe hemophilia A have FVIII activity below 1% of normal and experience spontaneous and uncontrolled bleedings that progressively cause arthropathies and may be fatal, if not properly treated[1]. Patient treatment is based on life-long prophylactic administration either of recombinant FVIII products that require at least weekly intravenous (i.v.) infusion to prevent hemorrhages, or of a recently approved activated-FVIII mimetic antibody that can be administered subcutaneously[2]. Despite the success of these drugs in improving clinical management and quality of life of people with hemophilia in high-income countries, gene therapy has long been considered a potentially definitive cure for hemophilia[3]. Advanced-phase clinical studies have highlighted the potential of gene therapy to fulfill this promise, by showing multiyear therapeutic benefit following a single i.v. administration of an adeno-associated virus (AAV) vector delivering a functional FIX or FVIII transgene to the liver, in adults affected by severe hemophilia B or A, respectively[4–6]. However, a decreasing trend in FVIII transgene expression has been reported, for reasons that are not fully understood and might be related to the challenge of stably maintaining functional episomal vector genomes reaching up to their packaging limit[6]. These studies represent milestones for gene therapy and provided the first evidence for safe and effective genetic modification of the human liver. However, AAV-vector gene therapy remains affected by some limitations: (i) the widespread pre-existing immunity to the parental virus, which precludes access to 20-30% of patients and imposes an immune-suppression regimen for a period of time following gene therapy to maintain AAV-transduced hepatocytes; (ii) dilution of episomal AAV vectors following liver growth; (iii) cargo capacity limited to 5 kb, particularly challenging for incorporating FVIII transgenes. HIV-derived lentiviral vectors (LV) may represent a complementary strategy for liver-directed gene therapy, for the following reasons: (i) low occurrence or feasible overcoming of immune barriers in humans, since worldwide prevalence of HIV infection is estimated at 0.8% and prior exposure to the vesicular stomatitis virus, whose surface glycoprotein (VSV-G) is used for pseudotyping LV, is rare, although natural low-titer antibodies (Abs) cross-reacting with the VSV-G protein are often present in human plasma; (ii) efficient integration of LV in the host cells' genome may be preferred for life-long maintenance of the therapeutic transgene and potentially allows treatment of pediatric patients without the need for vector re-administration[7]; (iii) larger packaging capacity may make LV better suited for transferring FVIII expression cassettes. Absence of prior clinical testing of systemic administration, manufacturing hurdles and concerns about insertional mutagenesis have until now hindered pre-clinical development of in vivo LV gene therapy directed to the liver. It has been shown that the natural source of most FVIII production is endothelial cells[8]. Gene transfer of LV expressing FVIII from liver endothelial cells has been proposed and some encouraging results have been reported in hemophilia A mice treated as adults[9,10]; however, the stability and turnover of these cells, both in post-natal liver growth and homeostasis in adulthood remain not fully understood[11]. Indeed, currently, the most clinically advanced AAV-based gene therapy strategies and the LV-based strategy described in this work exploit hepatocytes to produce transgenic FVIII[12].

We have previously shown that i.v. administration of LV results in efficient and long-term gene transfer to the liver and achieves phenotypic correction of hemophilia B in mice and dogs[13–15]. More recently, we generated allo-antigen free and phagocytosis-shielded (CD47hi) LV that allowed supra-normal activity of a human coagulation FIX transgene in non-human primates (NHP), without evidence of acute toxicity and clonal expansion of transduced cells[16,17]. Here we evaluated LV-mediated gene delivery of engineered versions of FVIII transgene in hemophilia A mice and in NHP, showing long-term FVIII activity and restoration of hemostasis, following i.v. administration to newborn and adult mice and normal-range human FVIII activity in NHP.

## Results

**LV expressing engineered FVIII transgenes allow phenotypic correction of hemophilia A mice**. We generated 3 different versions of human FVIII transgenes, all lacking the B domain, previously reported to be non-essential for clotting activity[18]: a wild-type (wt) sequence, a codon-usage optimized (co) and an engineered co version also containing an unstructured XTEN polypeptide in place of the B domain (Supplementary Fig. 1a)[19]. Inclusion of this polypeptide has been shown to increase protein stability and prolong circulating half-life of FVIII and other proteins, but, to our knowledge, has not been tested in a gene therapy setting[20]. We then cloned these transgenes (FVIII, coFVIII, coFVIII.XTEN) into a LV construct carrying a hepatocyte-specific enhancer promoter (Enhanced Transthyretin, ET) and target sequences for the hematopoietic-specific microRNA 142 (142 T). This LV design was previously shown to provide highly specific and robust transgene expression in hepatocytes[13,15,17,21]. Transduced mouse and human hepatic cell lines showed LV-dose-dependent FVIII production in the supernatant. The coFVIII and coFVIII-XTEN transgenes were expressed 18- and 22-fold higher than wt FVIII, respectively in the mouse cell line and 47- and 89-fold higher than wt FVIII, respectively in the human cell line, at matched LV DNA copies *per* cell (vector copy number, VCN; Supplementary Fig. 1b–g). We treated newborn (2-day old) hemophilia A (*F8* knock out, KO) mice by i.v. administration of $2.5 \times 10^{10}$ transducing units (TU)/kg of LV ($n = 5$–9 mice *per* LV) and measured FVIII blood concentration and activity overtime. We observed steady-state 19 ng/mL and 0.25 U/mL of FVIII corresponding to approximately 19 and 25% of normal in mice treated with LV.FVIII, while mice treated with LV.coFVIII or LV.coFVIII.XTEN produced 10-20-fold more FVIII corresponding to approximately 197 and 225 % of normal FVIII activity (Fig. 1a, b). Some of these mice developed anti-FVIII Abs starting 12 weeks post-LV (Fig. 1c); however, still maintaining circulating FVIII activity at the end of the experiment, despite a decreasing trend, likely due to Abs formation. All LV-treated mice, untreated hemophilia A and wt mice were then subjected to hemostatic challenge, by transecting the tip of the tail at the end of the experiment, 5–6 months post-LV and collecting the blood from the wound. Blood loss was then determined by measuring hemoglobin absorbance in the blood-containing solution. All the hemophilia A mice treated with LV.coFVIII or coFVIII.XTEN showed improved hemostasis, comparable to the range observed for wt mice, while hemostasis remained defective in mice treated with LV.FVIII that only expressed approximately 20% of normal FVIII on average (Fig. 1d). We measured LV VCN in different liver cell types at the end of the experiment, as previously described[17]. Average LV VCN in purified hepatocytes was between 0.2 and 0.54 (Fig. 1e). Average VCN in Kupffer cells (KC) and plasmacytoid dendritic cells was between 1 and 1.5, much lower than the VCN observed in mice treated by LV i.v. administration as adults, as we have previously reported[17] and show afterward in this work. FVIII output normalized on VCN in hepatocytes confirmed higher expression by coFVIII (32-fold compared to FVIII) with an additional nearly 2-fold increase by

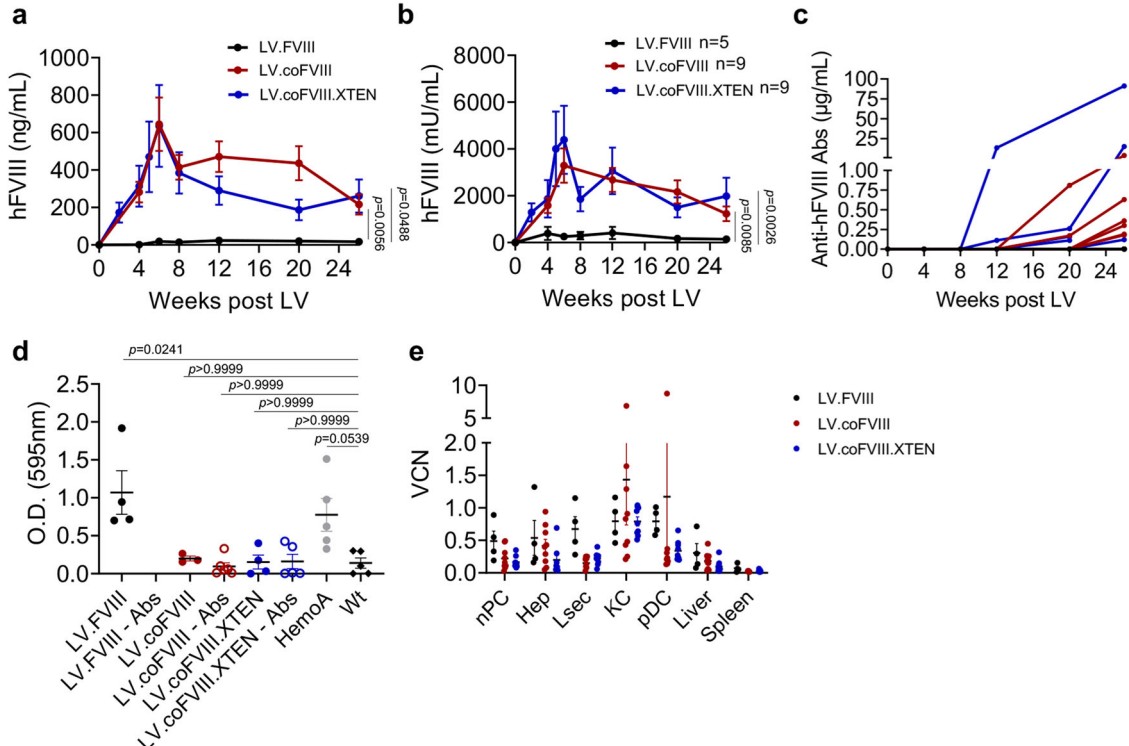

**Fig. 1 FVIII transgene selection in hemophilia A mice. a, b** Mean with standard error of the mean (SEM) of human FVIII (hFVIII) antigen (**a**) or activity (**b**) measured in the plasma of Hemophilia A (HemoA) mice treated as newborns by i.v. injection of $2.5 \times 10^{10}$ TU/kg of the indicated LV ($n = 5$, for LV.FVIII in black; $n = 9$ for LV.coFVIII in red; $n = 9$ for LV.coFVIII.XTEN in blue). Linear mixed model (LME) followed by post-hoc analysis (difference between experimental groups and the reference control group is evaluated at the last time point). Bonferroni's adjusted $p$-values from the *post-hoc* analysis are shown (**b**). **c** Single values of anti-hFVIII antibodies (Abs) measured in the plasma of mice in (**a, b**). **d** Single values and mean with SEM of hemoglobin absorbance at 595 nm collected during tail-clipping assay (see methods section) from LV-treated mice, HemoA untreated mice and wild type (wt) age-matched controls, as indicated. Abs-positive mice (empty dots) show hemostasis restoration similar to Abs-negative and wt mice. Kruskal–Wallis test followed by post-hoc analysis. **e** Single values and mean with SEM of vector copies *per* diploid genome (VCN) measured in fractionated and FACS-sorted liver subpopulations (nPC, non-parenchymal cells; Hep, hepatocytes; Lsec, liver sinusoidal endothelial cells; KC, Kupffer cells; pDC, plasmacytoid dendritic cells) or in total liver and spleen of mice in (**a, b**) 5–6 months after LV administration. O.D., optical density; co, codon-optimized; U: units. Source data are provided as a Source Data file.

FVIII.XTEN (Supplementary Fig. 2a). FVIII expression at steady-state well correlated with VCN in hepatocytes (Supplementary Fig. 2b), which was similar in both Abs-positive and Abs-negative mice (Supplementary Fig. 2c). Taken together, these data show improved protein output and therapeutic efficacy by the engineered FVIII transgenes.

**FVIII activity is maintained long-term following LV gene therapy in both newborn and adult hemophilia A mice.** We then treated additional neonatal hemophilia A mice and followed them for 1.5 years. Note that we reduced the dose of LV.coFVIII to $1 \times 10^{10}$ TU/kg, to obtain physiological-range FVIII activity. We showed FVIII blood concentration and activity that remained stable throughout the near lifetime follow-up at around 5–10% of normal on average (Fig. 2a, b). All these mice had low amounts of anti-FVIII Abs (Fig. 2c), with 3/4 maintaining stable FVIII activity and 1/4 showing a decreasing trend at the latest time points. VCN in purified hepatocytes was 0.02 at the end of the experiment (Fig. 2d), in line with FVIII expression. We then treated neonatal hemophilia A mice with increasing doses of LV.coFVIII ranging from $2.2 \times 10^9$ TU/kg to $3.5 \times 10^{10}$ TU/kg and detected circulating FVIII concentration and activity only at the 2 highest doses, which were on average 23 ng/mL, 276 mU/mL (corresponding to 23–28% of normal) at $1.7 \times 10^{10}$ TU/kg LV dose and 401 ng/mL, 7635 mU/mL (40–76% of normal) at

$3.5 \times 10^{10}$ TU/kg LV-dose (Fig. 2e, f). Note the almost 20-fold increase in FVIII transgene output despite the 2-fold increase in LV dose, in line with the data shown above. All treated mice but 1 had barely detectable anti-FVIII Abs (<1 µg/mL, Fig. 2g). We also performed an LV-dose–response of LV.coFVIII.XTEN in neonatal hemophilia and achieved between 5 and 82 ng/mL of circulating FVIII concentration and between 0.05 and 0.6 U/mL of FVIII activity (corresponding to 5 to 60–80% of normal) with an increasing LV dose–response relationship (Fig. 2h, i). The coFVIII.XTEN steady-state amounts were approximately between 2- and 7-fold higher than coFVIII at similar LV doses in these experiments (Fig. 2j). A smaller difference in FVIII amounts between the 2 transgenes was observed at the highest LV doses, in line with the previous experiments (see Fig. 1a, b and Supplementary Fig. 2a). FVIII output was stable for 6 months (longest time analyzed) except for mice that developed anti-FVIII Abs, starting from week 10 post-LV, in which circulating FVIII amounts declined as anti-FVIII Abs rose (Fig. 2k). Likely, the higher amounts of Abs in this experiment impaired FVIII detection whereas in the previous experiment lower amounts of Abs did not. Average VCN in the total liver ranged from 0.04 to 0.12 at the end of the experiment (Fig. 2l). To confirm increased half-life by the XTEN-containing FVIII, we administered recombinant B-domain deleted FVIII protein with or without XTEN i.v. to adult hemophilia A mice. We observed a slower decay of the XTENylated FVIII which remained 10-fold higher than FVIII at

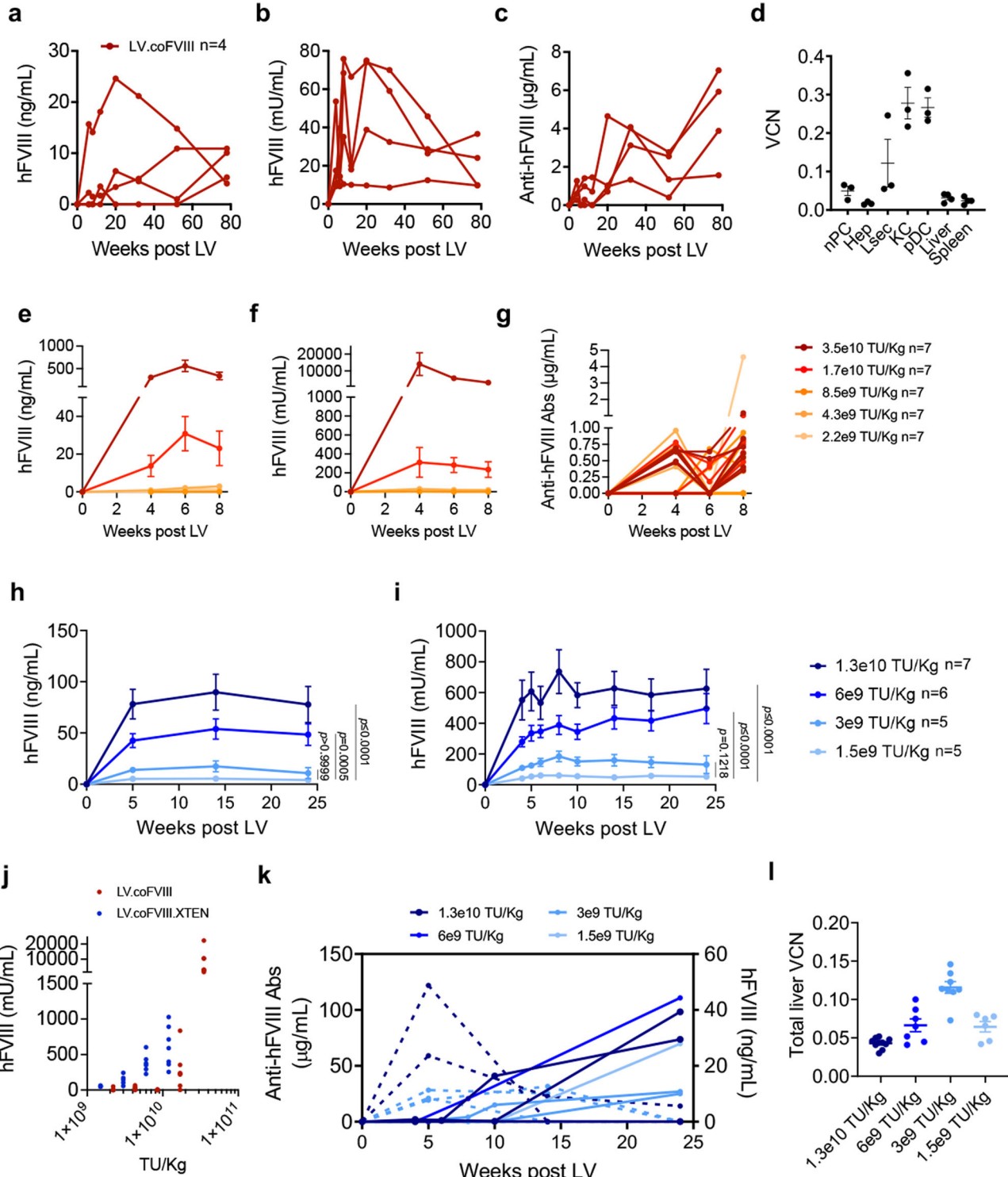

2 days after administration (Supplementary Fig. 2d). Taken together, these data show stable long-term FVIII transgene expression and activity by LV gene therapy from mice treated as newborns and improved steady-state FVIII output by the XTEN-carrying transgene, particularly at lower LV doses. In order to evaluate FVIII output and stability in adult mice, without confounding factors due to anti-FVIII immune responses, that are more likely to develop upon treatment at this age, we generated immune-deficient hemophilia A mice, by crossing *F8* KO with *Rag1* KO mice (RagHemoA). We confirmed that the double KO progeny was both devoid of circulating T and B lymphocytes and

showed prolonged clotting times (Supplementary Fig. 3). We then treated adult (8-week-old) RagHemoA mice by i.v. administration of $8 \times 10^{10}$ TU/kg of coFVIII.XTEN-containing LV. We observed an average concentration of 134 ng/mL and 1.12 U/mL activity of circulating FVIII in treated mice (corresponding to 134 and 112 % of normal), which remained stable for 11 months following gene therapy (Fig. 3a, b). Higher variability in FVIII output in adult compared to newborn mice may be due to lower permissiveness to transduction of hepatocytes in the former. The average VCN in purified hepatocytes at the end of the experiment was 0.37 (Fig. 3c). Notably, the average VCN in KC was 15, in line with our

**Fig. 2 Dose–response of LV expressing selected FVIII transgenes in newborn-treated HemoA mice monitored long-term. a–c** Single values of hFVIII antigen (**a**), activity (**b**), or anti-hFVIII Abs (**c**) measured in the plasma of HemoA mice treated as newborns ($n = 4$) by i.v. injection of $1 \times 10^{10}$ TU/kg of LV.coFVIII. **d** Single values and mean with SEM of VCN measured in fractionated and FACS-sorted liver subpopulations or in total liver and spleen of mice in (**a–c**) 20 months after LV administration. **e–g** Mean with SEM of hFVIII antigen (**e**), activity (**f**), or anti-hFVIII Abs (**g**) measured in the plasma of HemoA mice treated as newborns by i.v. injection of the indicated doses of LV.coFVIII ($n = 7$ per group). **h, i** Mean with SEM of hFVIII antigen (**h**), activity (**i**) measured in the plasma of Abs-negative HemoA mice treated as newborns by i.v. injection of the indicated doses of LV.coFVIII.XTEN ($n = 7$, for $1.3 \times 10^{10}$ TU/kg; $n = 6$, for $6 \times 10^9$ TU/kg; $n = 5$, for $3 \times 10^9$ TU/kg; $n = 5$, for $1.5 \times 10^9$ TU/kg). Linear mixed model (LME) followed by *post-hoc* analysis (difference between experimental groups and the reference control group is evaluated at the last time point). Bonferroni's adjusted *p*-values from the *post-hoc* analysis are shown. Due to the lower number of time points, to improve model fit, in panel (**h**), time variable has been entered in the model as categorical variable. In panel (**i**), four extreme outlier observations have been removed from the analysis. **j** Single values of plateau circulating hFVIII in mice shown in (**g**, red dots) and in (**i**, blue dots) at the indicated LV doses (for mice treated with LV.coFVIII.XTEN: $n = 7$, for $1.3 \times 10^{10}$ TU/kg; $n = 6$, for $6 \times 10^9$ TU/kg; $n = 5$, for $3 \times 10^9$ TU/kg; $n = 5$, for $1.5 \times 10^9$ TU/kg; for mice treated with LV.coFVIII: $n = 7$ for each dose). **k** Single values of anti-hFVIII abs (full lines, plotted on left *Y*-axis) and hFVIII antigen (dashed lines, plotted on right *Y*-axis) measured in the plasma of Abs-positive HemoA mice (3/10 mice treated at $1.3 \times 10^{10}$ TU/Kg; 2/7 mice treated at $6 \times 10^9$ TU/kg; 3/8 mice treated at $3 \times 10^9$ TU/kg; 1/6 mice treated at $1.5 \times 10^9$ TU/kg). **l** Single values and mean with SEM of VCN measured in the total liver of mice in (**h, i**) 6 months post-LV administration at the indicated doses. Source data are provided as a Source Data file.

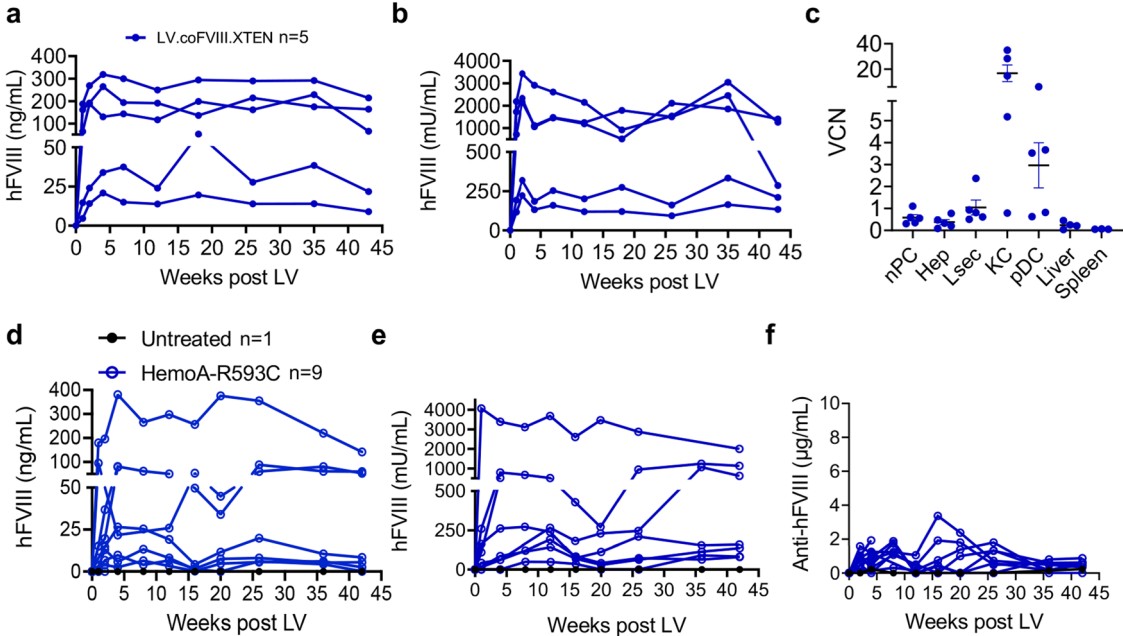

**Fig. 3 In vivo LV gene therapy in HemoA adult mice. a, b** Single values of hFVIII antigen (**a**) or activity (**b**) measured in the plasma of immunodeficient RagHemoA mice treated at 8 weeks of age ($n = 5$) by i.v. injection of $8 \times 10^{10}$ TU/kg of LV.coFVIII.XTEN. **c** Single values and mean with SEM of VCN measured in fractionated and FACS-sorted liver subpopulations or in total liver and spleen of mice in (**a, b**) 11 months after LV administration. **d** Single values of hFVIII antigen (**d**), activity (**e**) or anti-hFVIII Abs (**f**) measured in the plasma of immune-competent HemoA-R593C mice treated at 7 weeks of age ($n = 9$) by i.v. injection of $4 \times 10^{10}$ TU/kg of LV.coFVIII.XTEN.

previous data from mice treated as adults[17]. The lower VCN in KC observed at 6 months post-LV in mice treated when newborns (see Fig. 1e) suggest reduced KC transduction in newborn mice. We also evaluated LV gene therapy in an immune-competent hemophilia A mouse model, both *F8* KO and transgenic for a human FVIII point-mutant (HemoA-R593C). This mouse model completely lacks circulating FVIII protein and activity; however, it expresses a non-secreted human FVIII, thus it combines the hemophilia phenotype with immune tolerance to human FVIII epitopes, as previously described[22]. This mouse model better recapitulates hemophilia A patients previously treated with FVIII protein products and negative for anti-FVIII Abs that would be the target population for a first clinical testing of this gene therapy. I.v. administration of LV.coFVIII.XTEN in adult HemoA-R593C allowed long-term stable reconstitution of FVIII expression and activity on average 65% of normal (Fig. 3d, e) for 6 months after gene therapy. Treated mice presented minimal anti-FVIII Abs, which did not affect FVIII stability or activity (Fig. 3f). These data

show that LV gene therapy does not break immune tolerance in this immune-competent mouse model. Overall, these data show efficient transduction of hepatocytes and long-term stable fully normal reconstitution of FVIII activity in adult hemophilia A mice, by using the engineered codon-optimized XTEN-carrying FVIII transgene.

**In vivo LV gene therapy achieves normal-range human FVIII activity in NHP.** Collectively these data prompted us to evaluate LV-mediated human FVIII gene transfer in NHP. We thus produced large-scale purified batches of major-histocompatibility complex (MHC) free and CD47hi LV carrying either the coFVIII or coFVIII.XTEN transgene (Supplemental Table 1), by a good-manufacturing practice like process with a 2-step chromato-graphy purification, as previously reported[17]. We then treated 10 NHP with either LV at 2 different doses each: coFVIII at 3 or $6 \times 10^9$ TU/kg, coFVIII.XTEN at 1 or $3 \times 10^9$ TU/kg. Doses were

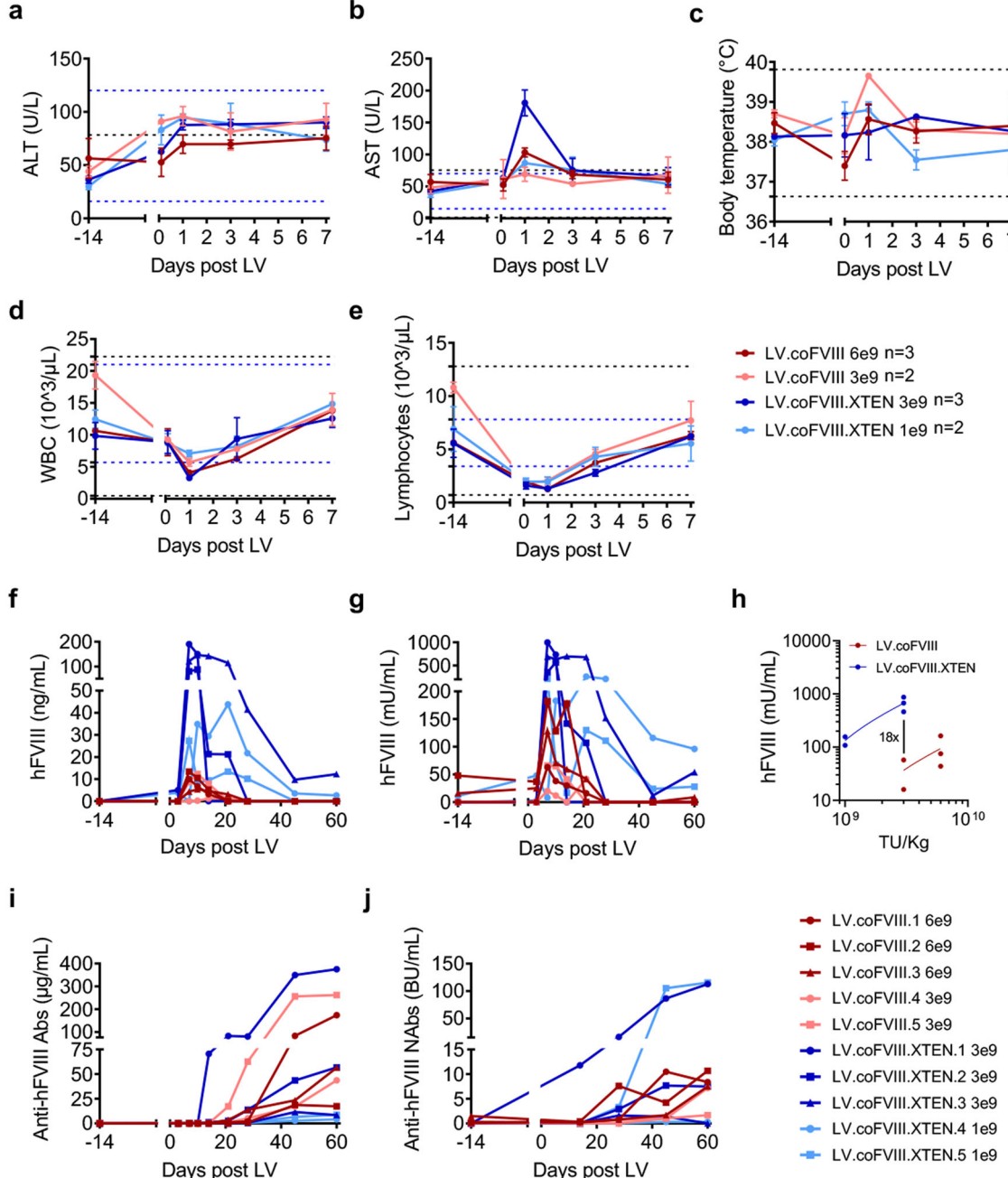

**Fig. 4 Tolerability and efficacy of in vivo LV gene therapy in NHP. a–e** Mean with SEM ($n = 3$) or range ($n = 2$) of the serum concentration of alanine aminotransferase (ALT, **a**), aspartate aminotransferase (AST, **b**), body temperature (**c**), counts of white blood cells (WBC, **d**) and lymphocytes (**e**) of NHP treated with LV.coFVIII ($n = 3$ $6 \times 10^9$ TU/kg; $n = 2$ $3 \times 10^9$ TU/kg), or with LV.coFVIII.XTEN ($n = 3$ $3 \times 10^9$ TU/kg; $n = 2$ $1 \times 10^9$ TU/kg) at the indicated time after administration. The black dashed lines show the mean±3 standard deviations (SD) calculated on a pool of 38 pre-LV samples taken from 19 animals; the blue dashed lines show the normal reference values for *Macaca fascicularis*. **f, g** Single values of the concentration of hFVIII antigen (**f**) or activity (**g**) measured in the plasma of LV-treated NHP (as indicated) at the indicated time after administration. **h** Mean of the concentration of hFVIII activity before Abs development in NHP treated as indicated at the indicated dose. **i, j** Single values of the concentration of total anti-hFVIII Abs (**i**), or neutralizing anti-hFVIII Abs (**j**) in the serum of LV-treated NHP (as indicated) at the indicated time after administration. BU: Bethesda Units.

selected based on previous experience with FIX-expressing LV[17]. Note that the LV-dose–response in NHP is more favorable than in mice, on a *per* weight basis, as previously described[17]. All LV-treated NHP received an immune suppressive regimen from 1 to 3 days before to 7-9 days after LV administration, to allow detection of human FVIII transgene in the absence of anti-FVIII Abs. We observed no major alteration of clinical or blood chemistry parameters, except for transient minor (1.2-2.6-fold

higher than the upper limit) elevation of serum aspartate aminotransferases (AST) and transient minor (1.7-2.4-fold lower than the lower limit) decrease of circulating lymphocytes (Fig. 4a–e), in line with our previous LV-FIX study. AST elevation positively correlated with the dose of LV particles infused (Supplementary Fig. 4a); however, it may also be related to muscle damage during animal handling for sample collection, since serum creatin phosphokinases were also elevated (Supplementary

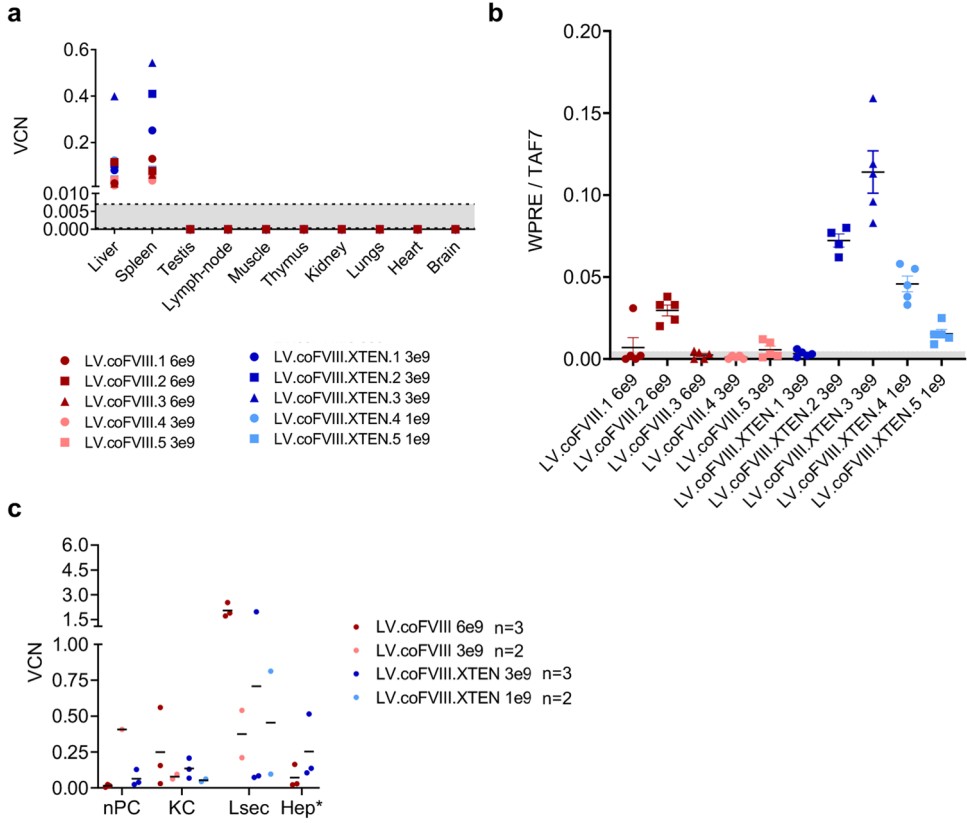

**Fig. 5 LV biodistribution in treated NHP. a** Single values of VCN in the indicated organs of LV-treated NHP, as indicated, at necropsy (60 days post-LV). The dashed lines defining the gray area represent the lower limit of detection (0.0003) and the lower limit of reliable quantification (0.007, see methods section). **b** Single values and mean with SEM of expression analysis by quantitative PCR of WPRE normalized on the endogenous TAF7 gene on RNA extracted from five different liver lobes of LV-treated NHP, as indicated. The gray area represents mean background signal for WPRE expression in the spleen. **c** Single values of VCN measured or calculated (*) in the indicated liver subpopulation after gradient purification (nPC) or FACS-sorting (KC, Lsec). VCN in Hep is calculated according to the formula $VCN_{liver} = (VCN_{Hep} \times 0.7) + (VCN_{nPC} \times 0.3)$. This formula is based on average proportion of liver cell subpopulations in the liver.

Tables 2–11). The blood concentration and activity of human FVIII peaked at 131 ng/mL and 0.68 U/mL (corresponding to 131 and 68% of normal) in NHP treated with LV.coFVIII.XTEN at $3 \times 10^9$ TU/kg, 22- and 17-fold higher than in NHP treated with LV.coFVIII at the same LV dose (Fig. 4f–h). NHP treated with the coFVIII.XTEN-carrying LV at $1 \times 10^9$ TU/kg showed 29 ng/mL and 0.2 U/mL of peak circulating FVIII concentration and activity, corresponding to 29 and 20% of normal, respectively. As expected, upon withdrawal of immune suppression, all NHP developed anti-FVIII Abs, though in different amounts, which in most cases were also neutralizing FVIII activity (Fig. 4i, j). Moreover, these Abs were cross-reacting with the endogenous NHP FVIII, because clotting times were prolonged (Supplementary Fig. 4b). Progressive reduction of hemoglobin and hematocrit, accompanied by increased in circulating reticulocytes in some cases, suggested the occurrence of bleeding events (Supplementary Fig. 4c–e). All other blood chemistry and hematology parameters mostly remained in the normal range throughout the follow-up (60 days post-LV), except for a few fluctuations (see Supplementary Tables 2–11). LV DNA was only detectable in the liver and spleen among the organs analyzed at the end of the experiment and ranged between 0.02 and 0.4 in the liver (Fig. 5a). Gene expression analysis in the liver at necropsy showed that the NHP treated with LV.coFVIII lost LV RNA (above background in only 1/5 animals), while LV.coFVIII.XTEN-treated NHP (4/5 animals) maintained it. These data are in line with remaining FVIII transgene at the end of the follow-up and suggest

maintenance of at least some transduced hepatocytes, despite induction of inhibitory FVIII Abs (Fig. 5b). Average LV VCN in KC purified from liver necropsies of treated NHP ranged between 0.05 and 0.25, showing effective protection of LV from phagocytosis by KC, as expected by the high surface content of CD47 on LV particles, and in line with our previous study[17] (Fig. 5c). Expression of LV RNA well correlated with both VCN in the total liver at the endpoint and with VCN in hepatocytes, calculated from the measured VCN in total liver and purified liver non-parenchymal cells (Supplementary Fig. 4f). Measurement of LV particles in the serum of treated NHP showed a very short LV circulating half-life, with 0.02-0.09% of administered particles remaining 1 day after infusion and becoming undetectable by day 3 (Supplementary Fig. 4g). Pathology analysis of liver and spleen showed no macroscopic or microscopic lesions, except for minimal hepatocytes atrophy with dilated sinusoids in one animal and mild neutrophilic infiltrate in the red pulp of the spleen in a second animal.

**NHP treated with the FVIII.XTEN transgene show reduced anti-FVIII immune responses**. We longitudinally monitored a panel of 26 cytokines and chemokines in treated NHP before and after LV administration to investigate acute innate response to LV particles and potentially detect delayed release related to developing adaptive responses against vector and/or transgene antigens (Ag). We observed that 10/26 analytes significantly increased in the first hours to days following LV administration and then

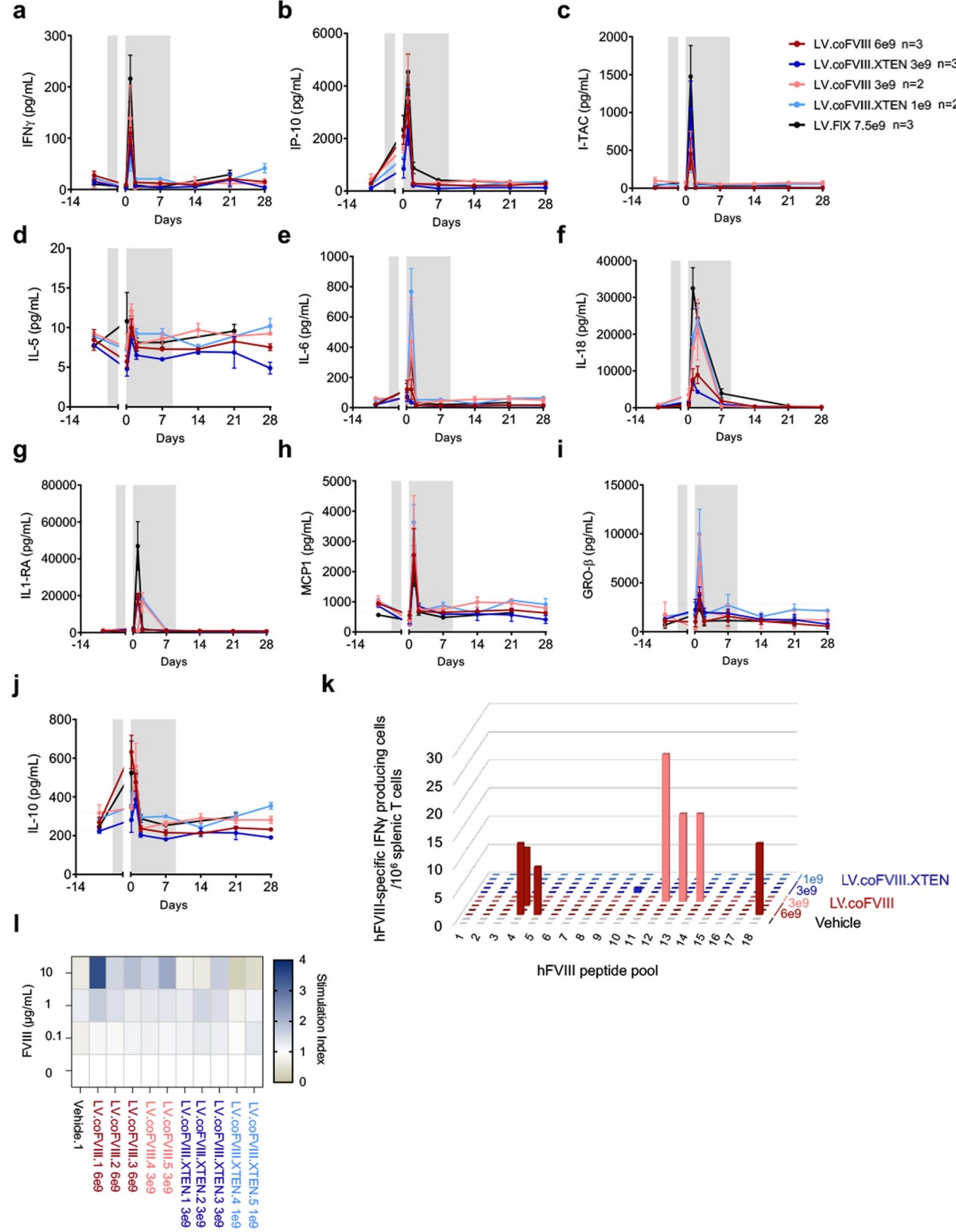

returned to baseline values in most cases already at day 2 post-LV (Fig. 6a–j). Similarly to the acute response previously reported following i.v. LV administration in murine models[17,23], pro-inflammatory cytokines, and chemokines, likely released from innate immune cells, significantly increased in the blood-stream of NHP after LV administration. We observed transient elevation of multiple signals for the recruitment and activation of innate immune effectors, such as neutrophils, eosinophils, monocytes, and dendritic cells, and in particular growth-regulated protein beta (GRO-β), interleukin (IL)-5, monocyte chemoattractant protein 1 (MCP1), IFNγ and interferon gamma-induced protein 10 (IP-10) within the first 1–2 days post-LV, as well as

**Fig. 6 Immune responses following in vivo LV gene therapy in NHP. a–j** Mean and SEM ($n = 3$, NHP treated with $6 \times 10^9$ TU/kg of LV.coFVIII or with $3 \times 10^9$ TU/kg of LV.coFVIII.XTEN) or range ($n = 2$, NHP treated with $3 \times 10^9$ TU/kg of LV.coFVIII or with $1 \times 10^9$ TU/kg of LV.coFVIII.XTEN) of the serum concentration of interferon-γ (IFNγ, **a**), interferon gamma-induced protein 10 (IP-10, **b**), interferon–inducible T-cell alpha chemoattractant (I-TAC, **c**), interleukin-5 (IL-5, **d**), IL-6 (**e**), IL-18 (**f**), IL-1 receptor antagonist (IL-1RA, **g**), monocyte chemoattractant protein 1 (MCP1, **h**), growth-regulated protein beta (GRO-β, **i**) and IL-10 (**j**) in LV-treated NHP, as indicated. Black dotted lines are mean±1 SD calculated on a pool of pre-LV samples taken from 20 animals. Gray area represent the window of immune-suppression regimen in LV.coFVIII or LV.coFVIII.XTEN-treated NHP (see "methods" section for details). **k** Histogram reporting the frequency of hFVIII-specific IFNγ producing T cells/$10^6$ T cells in NHP treated as indicated. The 18 pools of FVIII peptides cover the entire transgenes encoded by the administered LV (see Supplementary Fig. 7 for ELISpot wells). **l** Heatmap reporting the stimulation index (*i.e.* proliferation over unstimulated cells) of splenocytes kept in culture in presence of the indicated amounts of hFVIII, as indicated. Source data are provided as a Source Data file.

interferon–inducible T-cell alpha chemoattractant (I-TAC) and MCP1 chemokines for the recruitment and activation (IL-6) of T and B lymphocytes. By comparing the peak concentration of each of these cytokines among the different groups of NHP, we did not find any general increase at increasing LV doses (Supplementary Fig. 5a–j). Moreover, we repeated the measurement of these cytokines on serum samples from non-immune suppressed NHP treated with LV encoding human FIX from our previous study and observed that the peak concentration of these cytokines was similar to that observed in NHP treated in this study, suggesting that the administered immune suppression did not substantially alter the innate immune response to LV administration (see Supplementary Fig. 5a–j). Interestingly, we noted that the peak concentration of some of these cytokines seemed to inversely correlate with the administered LV particle dose (Supplementary Fig. 5k–m). This data might suggest that increasing CD47 signaling on professional phagocytes from high amounts of infused vector particles reduces cytokine release. We observed a rise followed by a decline of anti-VSV-G Abs, as expected due to i.v. administration of VSV-G pseudotyped LV (Supplementary Fig. 6a). However, in some NHP the concentration of these Abs was lower than in previous reports[17], suggesting that the immune-suppression regimen reduced the humoral immune response to this Ag that was only present at the time of LV administration. In parallel to humoral response to FVIII and VSV-G, anti-FVIII-specific T lymphocytes were quantified over time in peripheral blood mononuclear cells (PBMC) and at the end of the experiment in splenocytes, by enumerating IFNγ-producing T cells in response to 18 peptides pools covering the entire FVIII aminoacidic sequence. We observed low-frequency sporadic anti-FVIII T-cell responses in PBMC in all treatment groups (Supplementary Fig. 6b). Analyses of secondary lymphoid organs revealed that FVIII-responsive T cells were not detectable in hepatic lymph nodes (Supplementary Fig. 6c). Conversely, IFNγ-producing T cells were detected in response to FVIII stimulation in splenocytes of NHP treated with LV.coFVIII (3/5), while only a weak response was present in 1/5 NHP treated with LV.coFVIII.XTEN (Fig. 6k and Supplementary Fig. 7). This rapid in vitro response upon FVIII peptide stimulation (24 h of stimulation) correlates with the lower residual transgene expression in the liver (see Fig. 5b), therefore likely derived from memory FVIII-specific cytotoxic T cells that mediated elimination of FVIII-expressing hepatocytes. In line with these results, splenic T cells from NHP treated with LV.coFVIII displayed a higher proliferative response upon in vitro culture in the presence of increasing concentration of FVIII protein, which suggests that a more pronounced T-cell response was mounted in NHP to LV.coFVIII treatment (Fig. 6l). Splenocytes were also tested to evaluate the presence of IL-5 producing T helper cells, in response to FVIII peptides stimulation, which however were undetectable in all NHP at necropsy (Supplementary Fig. 6d). Positive controls, T-cell frequencies and pictures of wells for spleen enzyme-linked immunospot (ELISpot) assays are reported in Supplementary

Fig. 7. In order to better visualize and correlate data of innate and adaptive immune responses and data of FVIII expression at peak and endpoint, we generated a heatmap of these results (Supplementary Fig. 8). We noted that amounts of remaining FVIII transgene activity and RNA at endpoint inversely correlated with the presence of FVIII-reactive T cells in the spleen. Moreover, higher cytokine response did not appear to predict higher amounts of FVIII-specific T cells at endpoint, rather correlates with higher initial FVIII output. Collectively, these data show that the applied immune suppressive regimen minimally affected the innate immune responses, which likely still recruited and activated subsets of antigen-presenting cells and lymphocytes in the spleen and in the liver, leading to the induction of adaptive anti-FVIII immune responses once the immunosuppression was withdrawn. Overall, these results indicate that the NHP treated with FVIII.XTEN encoding LV expressed the highest FVIII amounts despite the lowest administered LV doses and showed the lowest anti-FVIII cellular immune response, while maintaining FVIII-expressing hepatocytes at the end of the follow-up.

## Discussion

Here we show that liver-directed LV gene therapy allows establishing long-term FVIII activity in the blood and phenotypic correction of hemophilia A mice and achieves therapeutic-range human FVIII activity in NHP, albeit only during immunosuppression. The codon-optimization of the B-domain deleted FVIII transgene provided for 10-20-fold increase in expression both in human hepatic cell lines and in mice, in line with previous reports[24,25]. Incorporation of the unstructured XTEN polypeptide into the FVIII transgene further improved the steady-state level and activity of the protein in vivo. XTENylation of biologically active molecules has been shown to increase protein half-life in pre-clinical models and in humans, by increasing the hydrodynamic volume of the modified molecule[20]. A recombinant XTEN-containing FVIII product has shown 3-4-fold half-life extension compared to conventional FVIII in humans, without evidence of adverse events or increased incidence of development of neutralizing anti-FVIII Abs[26]. Beyond these data, here we show a remarkable 10-fold higher and 20-fold higher FVIII transgene expression and activity in mice and NHP, respectively, by inclusion of the XTEN element, thus significantly reducing the LV dose required for therapeutic efficacy, decreasing manufacturing needs and alleviating concerns related to possible LV-dose-dependent toxicities. These results may be due to increased half-life of the protein in the circulation, as shown here, especially at physiological blood concentrations, and potentially to improved FVIII secretion by hepatocytes, thus overall resulting in higher plateau blood FVIII concentration and activity. The difference between mice and NHP may be due to the different LV doses used and resulting FVIII amounts in the circulation of LV-treated animals. We consider unlikely that the anti-human FVIII immune response observed in NHP upon corticosteroid removal,

influenced peak amounts of circulating human FVIII in the NHP, since the animals were still immune suppressed at that time.

Long-term stability of FVIII-XTEN transgene expression and activity in both mice treated as newborns and adults indicates the absence of selective disadvantage of transduced hepatocytes, which may be due to toxicity from FVIII overexpression. On the same line, the absence of increase in transgene over time indicates a lack of expansion of transduced hepatocytes, which may potentially be triggered by LV insertional mutagenesis. These data also confirm that LV gene therapy allows maintenance of the therapeutic transgene following liver growth and homeostasis in mice. In humans, restoration of stable, therapeutic amounts of FVIII activity since early childhood may be preferred to avoid the accumulation of joint damage, by preventing bleeding as early in life as possible[27]. Interestingly, LV-dose–response was more favorable in mice treated as newborns compared to mice treated as adults, suggesting better access and/or permissiveness to LV transduction in the former, a finding that warrants further investigation. We also report the generation of novel double KO mice that combine the hemophilia and immune-deficient phenotypes, a useful mouse model for pre-clinical investigation of gene transfer and editing strategies or FVIII protein products.

In agreement with our previous LV-FIX study, the present study confirms minor and self-resolving acute toxicities and pro-inflammatory response, following i.v. administration of LV in NHP, across different LV doses and transgenes. The corticosteroid regimen administered for a few days around LV infusion temporarily inhibited the adaptive arm of the immune system, thus allowing the determination of peak FVIII transgene expression and activity for a window of time, before the development of anti-FVIII humoral and cellular immune responses, which occurred upon corticosteroid removal. However, this immune-suppression regimen did not apparently affect the innate immune response to LV administration, as shown from the similar cytokine response observed in NHP treated in this study compared to NHP treated in our previous study, which received i.v. LV-FIX administration without corticosteroid treatment[17]. The lack of increase in cytokine release when increasing the amount of infused LV particles suggests that the high surface content of CD47 *per* vector particle conferred shielding from uptake and innate immune sensing also in a dose-dependent non-particle autonomous manner. We also confirm selective targeting of liver and spleen of CD47hi LV and protection from phagocytosis by KC in NHP, following peripheral vein infusion.

Restoration of FVIII activity above 12% of normal in people with severe hemophilia A is considered adequate to mostly prevent joint hemorrhages and may thus be set as the minimal target for correction in gene therapy[28]. The favorable acute toxicity profile observed, combined with therapeutic-range LV dose-dependent FVIII activity, provide a comfortable therapeutic window and encourage further development of LV-mediated liver gene therapy for hemophilia A. Because of the high immunogenicity of the FVIII protein, anti-human FVIII immune responses are expected in NHP and have been previously observed in pre-clinical studies of both FVIII protein and AAV-vector mediated gene replacement[25,29]. The absence of anti-FVIII Abs development following AAV-vector gene therapy in human trials in Abs-negative people previously treated with FVIII protein replacement therapy suggests that this population is at low risk of mounting immune responses to FVIII following gene therapy and that the value of the NHP model is limited in predicting this outcome. Maintenance of FVIII activity following LV gene therapy in immune-competent hemophilia A mice transgenic for the R593C human FVIII mutant supports this hypothesis. However, further investigation and possible modulation of anti-FVIII immune responses in pre-clinical models is anticipated to further reduce the risk of inducing such responses in humans and in view of future application of LV gene therapy to previously untreated hemophilia patients that are naïve to FVIII. With this perspective, the inclusion of the XTEN polypeptide in the FVIII transgene might alleviate some of its immunogenicity, as the NHP treated with FVIII.XTEN LV showed the highest FVIII transgene RNA in the liver and circulating activity at the end of the follow-up, accompanied by a reduced FVIII-specific T-cell response compared to NHP treated with LV expressing the non-modified transgene. The overall low frequency of anti-FVIII T-cell responses detected in this study suggests that more sensitive immune assays are needed to confirm these data and further assess anti-transgene immune responses in future studies carried out in similar experimental settings.

Integration of LV in the genome of target cells is required to maintain the therapeutic transgene in dividing cells, while raising concerns of inducing insertional mutagenesis. However, accumulating evidence supports the low genotoxicity of LV, as no insertional oncogenesis has been published so far in >300 patients treated by LV-transduced hematopoietic stem cells across different doses and transgenes, with multiyear follow-up and mostly at high VCN[30], although clonal expansions leading in 2 patients to myelodysplastic syndrome were recently reported in a single trial using LV with strong viral enhancer/promoter (https://doi.org/10.1089/hum.2021.29180.abstracts). Moreover, we have previously shown undetectable genotoxicity in the liver by our LV design in sensitive mouse models, dogs, and NHP[15,17,31]. Whereas efforts are underway to continue modeling and possibly reducing LV genotoxic potential, the safety track records of current LV design in ex vivo gene therapy in humans reassures about their potential use for in vivo gene therapy.

In conclusion, surface-engineered MHC-free and CD47hi LV have demonstrated efficient gene transfer into hepatocytes in NHP. By engineering the FVIII transgene we further enhanced the therapeutic index of in vivo LV gene therapy for hemophilia A, paving the way for potential clinical application.

## Methods

**Study design**. The sample size in experiments with mice was chosen according to previous experience with experimental models and assays. The sample size in the NHP study was limited by ethical and feasibility reasons. No sample or animal was excluded from the analyses. Mice and NHP were randomly assigned to each experimental group. Investigators were not blinded.

**Vector production**. For the NHP study, we used large-scale purified CD47hi and MHC-free LV batches, produced by MolMed S.p.A. (now AGC Biologics), on 24 L scale of supernatant and formulated in PBS 0.2% human serum albumin. The vector batches were produced by using a large-scale validated process and following pre-GMP guidelines[32]. Briefly, LV is produced by transient 4 plasmid transfection of CD47hi MHC-negative 293T cells in 10-tray cell factories by calcium phosphate precipitation. Twenty-four hours after removal of the transfection medium, the cell supernatant is harvested and stored at 4 °C. The culture medium is replaced and after a further 24 h a second harvest is performed. The medium collected from the two harvests is pooled and filtered through 5/0.45 μm filters to discard cell debris. The downstream purification process includes a benzonase treatment overnight at 4 °C, followed by a DiEthylAmino Anion Exchange (DEAE) chromatography step, concentration, and gel filtration in PBS or PBS 5% dimethyl sulfoxide (DMSO). The resulting LV preparation undergoes 0.2 or 0.45 μm filtration and aseptic filling. The purified vector preparation is stored at −80 °C. Results of selected quality control assays performed on these batches are reported in Supplemental Table 1. For all the other experiments in vitro and with mice, we used lab-grade LV. Lab-grade third-generation self-inactivating (SIN) LV were produced by calcium phosphate transient transfection into 293T cells (from ATCC). 293T cells were transfected with a solution containing a mix of the selected LV genome transfer plasmid, the packaging plasmids pMDLg/pRRE and pCMV.REV, pMD2.VSV-G and pAdvantage (Promega). Calcium phosphate-mediated transfection: $9 \times 10^6$ 293T cells are seeded 24 h before transfection in 15-cm dishes. Two hours before transfection culture medium is replaced with a fresh medium. For each dish, a solution containing a mix of the selected transfer plasmid, the packaging plasmids pMDLg/pRRE and pCMV.REV, pMD2.G and the pAdvantage plasmid is prepared using 35, 12.5, 6.25, 9, and 15 μg of plasmid DNA, respectively. A 0.1X TE solution

(10 mM Tris-HCl, 1 mM EDTA pH 8.0 in dH20) and water (1:2) are added to the DNA mix to 1250 µl of the final volume. The solution is left on a spinning wheel for 20–30 min, then 125 µl of 2.5 M CaCl₂ are added. Right before transfection, a precipitate is formed by adding 1250 µl of 2X HBS (281 mM NaCl, 100 mM HEPES, 1.5 mM Na₂HPO₄, pH 7.12) while the solution is kept in agitation on a vortex. The precipitate is immediately added to the culture medium and left on cells for 14-16 h and after that the culture medium is changed. The supernatant is collected 30 h after the medium change and passed through a 0.22 µm filter (Millipore). Filtered supernatant is transferred into sterile 25 × 89 mm poliallomer tubes (Beckman) and centrifuged at 20,000 $g$ for 120 min at 20 °C (Beckman Optima XL-100K Ultracentrifuge). Vector pellet is dissolved in the appropriate volume of PBS to allow a 500X or 1000X concentration.

**Mice experiments**. Founder B6;129S-*F8*tm1Kaz/J mice (referred to as HemoA or *F8* KO[33]) were obtained from The Jackson Laboratories (stock #004424). Founder B6.129S7-Rag.1t1Mom/J mice (referred to as *Rag1* KO[34]) were purchased from The Jackson Laboratory (stock # 002216). *F8-Rag1* double KO (referred to as RagHe-moA) mice were obtained by crossing *F8* KO homozygous mice with *Rag1* KO homozygous mice. Founder *F8*tm1Kaz Tg(Alb-F8*R593C)T4Mcal/J mice (referred to as HemoA-R593C[22]) were obtained from The Jackson Laboratories (stock #017706). For mice genotyping, DNA was extracted from tail biopsies using Maxwell 16 Mouse Tail DNA Purification Kit (Promega), following the manufacturer's instructions. The genotype was then assessed following the protocols available on The Jackson Laboratory website. All mice were maintained in specific pathogen-free conditions and fed *ad libitum* with VRF1 (P) by Special Diet Services. Vector administration was carried out in males and females adult (7–10-week-old) mice by tail-vein injection (250–500 µL/mouse), while in newborns by temporal vein injection (25–30 µL/mouse). Mice were bled from the retro-orbital plexus using capillary tubes and blood was collected into 0.38% sodium citrate buffer, pH 7.4. Mice were deeply anesthetized with tribromoethanol (Avertin) and euthanized by CO₂ inhalation at the scheduled times. All animal procedures were approved by the Institutional Animal Care and Use Committee (#757, #978) and the national government. The study complies to all relevant ethical regulations related to the use of research animals.

**Coagulation assays**. The tail-clipping assay was performed as described[14]. Mice were anesthetized and tail was placed in pre-warmed 37 °C water for 2 min and subsequently cut at 2.5–3 mm diameter. Tail was then immediately placed in 37 °C PBS with calcium and magnesium and monitored for bleeding or clotting for 15 min. Blood-containing PBS was centrifuged at 520 $g$ for 10 min at 4 °C to collect erythrocytes and re-suspended in 6 ml of lysis buffer (10 mM KHCO3, 150 mM NH4Cl, 0.1 mM EDTA). Lysis proceeded for 10 min at room temperature and samples were centrifuged as above. OD at 595 nm of supernatants was measured.

To confirm RagHemoA phenotype, the clotting time of mouse plasma was determined by activated partial thromboplastin time (aPTT) using a semiautomated coagulometer (BioMerieux, France).

**FVIII assays**. The concentration of human FVIII was determined in mouse plasma by an enzyme-linked immunosorbent assay (ELISA) specific for human FVIII antigen. Microtiter plates were coated with anti-hFVIII binding Ab (Green Mountain Antibodies #GMA8016, 0.2 µg/well in 0.1 M carbonate buffer, pH 9.6) over night at 4 °C and then blocked 1 h at room temperature with blocking buffer (PBS 0.05% Tween-20, 1 M NaCl, 10% heat inactivated horse serum, Gibco). Plasma samples are diluted as needed starting from 1:10 in blocking buffer, added to wells (100 µL/well) and incubated 2 h at 37 °C. hFVIII was detected by adding detection Ab (Affinity Biologicals, F8C-EIC-D) 1 h at 37 °C, followed by 5–10 min incubation with 100 µL/well of TMB substrate (Surmodics). The reaction was blocked with HCl 1 N (50 µL/well) and absorbance of each sample was determined spectrophotometrically at 450 nm, using a Multiskan GO microplate reader (Thermo Fisher Scientific) and normalized to antigen standard curve (ReFACTO, Pfizer, from 25 ng/mL to 0.39 ng/mL serially diluted 1:2 in blocking buffer; dilution was corrected with 10% HemoA murine plasma). hFVIII activity in mouse plasma was measured using Coatest SP FVIII (Chromogenix) following the manufacturer's instructions.

Anti-hFVIII Abs were measured in mouse plasma by ELISA. Microtiter plates were coated with ReFACTO (Pfizer, 0.1 µg/well in 0.1 M carbonate buffer, pH 9.6) over night at 4 °C and then blocked 1 h at room temperature with blocking buffer (PBS 0.05% Tween-20, 10% heat inactivated horse serum, Gibco). Samples are heat inactivated for 30 min at 56 °C, diluted as needed starting from 1:100 in blocking buffer, added to wells (100 µL/well) and incubated 2 h at 37 °C on orbital shaker. Anti-hFVIII Abs were detected by adding detection Ab (goat anti-mouse IgG-HRP, Sigma, 1:10,000 in blocking buffer) 1 h at 37 °C on orbital shaker, followed by 5-10 min incubation with 100 µL/well of TMB substrate (Surmodics). The reaction was blocked with HCl 1 N (50 µL/well) and absorbance of each sample was determined spectrophotometrically at 450 nm, using a Multiskan GO microplate reader (Thermo Fisher Scientific) and normalized to the standard curve. The standard curve is a pool of 7 different commercial anti-human FVIII Abs raised against different FVIII domains (Green Mountain Antibodies #GMA8002, #GMA8005, #GMA8008, #GMA8011, #GMA8015, #GMA8016, #QED10104)

serially diluted 1:2 from 100 ng/mL to 0.78 ng/mL in blocking buffer; dilution was corrected with 1% HemoA murine plasma.

The concentration of hFVIII was determined in NHP plasma by an ELISA specific for hFVIII antigen. Microtiter plates were coated with anti-hFVIII binding Ab (Green Mountain Antibodies #GMA8023, 0.2 µg/well in 0.05 M carbonate buffer, Sigma) over night at 4 °C and then blocked 1 h at room temperature with blocking buffer (PBS 0.05% Tween-20, 0.5 M NaCl, 10% heat inactivated horse serum, Gibco). Plasma samples are diluted as needed starting from 1:10 in blocking buffer, added to wells (100 µL/well) and incubated 2 h at 37 °C. hFVIII was detected by adding detection Ab (Affinity Biologicals, F8C-EIA-D) 1 h at 37 °C, followed by addition of 100 µL/well of TMB substrate (Surmodics). Reaction was blocked with HCl 1 N (50 µL/well) and absorbance of each sample was determined spectrophotometrically at 450 nm, using a Multiskan GO microplate reader (Thermo Fisher Scientific) and normalized to antigen standard curve (ReFACTO, Pfizer or recombinant hFVIII-XTEN, concentrated from supernatant of transfected 293 cells (see also Supplementary materials and methods), from 50 ng/mL to 0.39 ng/mL serially diluted 1:2 in blocking buffer; dilution was corrected with 10% NHP plasma). hFVIII activity was quantified in NHP plasma by a modified FVIII chromogenic assay: hFVIII in plasma samples was first captured by a hFVIII-specific Ab immobilized onto a 96-well plate (Green Mountain Antibodies, #GMA8023, 0.1 µg/well in 0.05 M carbonate buffer, Sigma), then chromogenic activity of hFVIII was measured using Coatest SP FVIII kit (Diapharma, K824086) following the manufacturer's instructions. Standard curves were obtained by diluting recombinant human FVIII (ReFACTO, Pfizer) into untreated NHP plasma. Total anti-human FVIII Abs were quantified in heat inactivated NHP serum (1 h at 56 °C) by ELISA. Microtiter plates were coated with ReFACTO (Pfizer, 0.1 µg/well in 0.1 M carbonate buffer, pH 9.6) over night at 4 °C and then blocked 1 h at room temperature with blocking buffer (PBS 0.05% Tween-20, 0.5 M NaCl, 10% heat inactivated horse serum, Gibco). Samples were diluted as needed starting from 1:20 in blocking buffer, added to wells (100 µL/well) and incubated 2 h at 37 °C on orbital shaker. Anti-hFVIII Abs were detected by adding detection Ab (rabbit anti-monkey IgG-HRP, Sigma #A2054, 1:10,000 in blocking buffer) 1 h at 37 °C on orbital shaker, followed by 5-10 min incubation with 100 µL/well of TMB substrate (SurModics). The reaction was blocked with HCl 1 N (50 µL/well) and absorbance of each sample was determined spectrophotometrically at 450 nm, using a Multiskan GO microplate reader (Thermo Fisher Scientific) and normalized to the standard curve (Monkey IgG, MyBioSource #MBS679190, from 125 ng/mL to 1.9 ng/mL serially diluted 1:2 in coating buffer). Neutralizing anti-human FVIII Abs were determined in heat inactivated NHP plasma samples (1 h at 56 °C) by Bethesda assay. The test sample was incubated for 2 h at 37 °C with recombinant human FVIII (ReFACTO, Pfizer, 1 U/mL) with FVIII activity of 100%. Residual FVIII activity was measured using Coatest FVIII SP kit (Diapharma, K824086) and converted into Bethesda Units (BU)/mL, where one BU is defined as the inverse of the dilution factor of the test sample that yields 50% residual FVIII activity. The cutoff value is 1 BU/mL, calculated on mean + 2 standard deviations on 10 pre-gene therapy samples.

**NHP study**. Ten adults (3–5 kg body weight) males *Macaca Leonina* (Northern pig-tailed macaques) were purchased by Bioprim (Baziège, France). Macaques were housed in an enriched environment with access to toys, fresh fruits, and vegetables at the Boisbonne Center (Nantes, France), under protocol APAFIS#4302-2015122314563838 that was approved by the Institutional Animal Care and Use Committee of the Pays De Loire. The study complies to all relevant ethical regulations related to the use of research animals. For LV administration, animals were anesthetized with Demetomidine (Domitor) 30 µg/kg and Ketamine 7 mg/kg and maintained under gas anesthesia, 1-2% Isoflurane (Vetflurane). The LV-containing solution was administered using a syringe with controlled flow rate fixed at 1.5 mL/min via a catheter in the saphenous vein (31–100 total mL according to LV type and dose, 6.6–23.5 mL/kg). Each NHP was treated with pools coming from different LV batches (see Table S1). Solu-Medrol (methylpredniso-lone, 10 mg/kg/day) was administered intramuscularly as immune-suppression regimen from day −1 to day 7 for high dose LV.coFVIII and LV.coFVIII.XTEN-treated NHP, while it was extended from day −3 to day 9 for low dose treated NHP (LV administration occurred at day 0). In addition, an antihistamine pre-medication regimen was administered: dexchlorpheniramine (Polaramine, 4 mg/kg) i.v. 30 min before LV. Blood samples were taken at different time points from the femoral vein upon anesthesia with 10 mg/kg ketamine (Imalgene) intramuscularly. For hematology, 1 mL of total blood samples was collected on EDTA-coated tubes. For clinical biochemistry, 2 mL of total blood samples were collected on heparin-coated tubes and for hemostasis, 1.8 mL of total blood was collected in citrate-coated tubes. Blood tests were performed on fresh samples at the Veterinary School of Nantes (LDHvet, Oniris). Biochemistry parameters were analyzed by automatons and analyzers based on spectrometry, reflectometry, potentiometry, and enzyme immunoassays. Hemostasis was analyzed by a hemostasis analyzer based on electro-mechanical clot detection (viscosity-based detection system). Tissue samples were collected following euthanasia, performed by i.v. injection of pentobarbital sodium (Dolethal).

**ELISpot splenocytes/PBMC**. PBMC were isolated by density gradient (Ficoll, Sigma-Aldrich), splenocytes, and hepatic LN cells were obtained by disruption of the tissue and cryopreserved. Multiscreen filter plate (Millipore-Merck) were

coated overnight 4 °C with capture Ab for interferon-γ (IFNγ, MT126L 10 μg/mL 50 μL/well, Mabtech) or IL5 (TRFK5 10 μg/mL 50 μL/well, Mabtech) and blocked with PBS 1% BSA for 2 h at 37 °C. Plates were equilibrated with culture medium for 10 min at room temperature before seeding cells. Total NHP splenocytes ($3 \times 10^5$ cells/well), LN cells or PBMC ($2.5 \times 10^5$ cells/well) were plated in X-vivo-15 (Lonza) at least in duplicates without antigenic stimulation (DMSO alone) or stimulated with 18 different peptide pools covering the entire aminoacidic sequence of FVIII.XTEN (each peptide at 1 μM final concentration, peptide library 15aa long, 5aa offset, Sigma-Aldrich) for 24 h at 37 °C 5% $CO_2$. Tetradecanoyl phorbol acetate and Ionomycin (TPA/iono) stimulation served as positive control as polyclonal inducer of cytokine release. At the end of the culture, detection Ab for IFNγ (7-B6-1 1 μg/mL 50 μL/well, Mabtech) or IL-5 (5A10 1 μg/mL 50 μL/well, Mabtech) was added and incubated for 2 h at room temperature. Avidind-POD solution (Roche, 1:5,000, 50 μL/well) was then added and incubated for 1 h at room temperature and spot were developed by AEC solution (Sigma-Aldrich) for 15 min at room temperature in the dark. A plate image was acquired and spots were counted by Immunospot S6-Ultra (Cellular Technology Limited). The mean number of spots+2 SD from the unstimulated condition was subtracted to stimulated condition and reported as IFNγ or IL-5 producing cells in $1 \times 10^6$ splenic T cells or PBMC or LN cells. The number of splenic T cells/well has been calculated based on cytofluorimetric analysis of cell before seeding (anti-human CD4 clone MT477, and anti-human CD8 clone SK1 cross-reacting with Macaca Leonina).

**Splenocyte proliferation.** Total NHP splenocytes were plated in flat-bottom 96-well plate ($3 \times 10^5$ cells/well) in X-vivo-15 (Lonza). Cells were left unstimulated or stimulated with increasing dose of FVIII.BDD protein (ReFACTO, Pfizer) in triplicates. Anti-NHP CD3 (clone CD3-1) stimulation served as positive control as polyclonal inducer of T-cell proliferation. After 5 days of culture, 1μCi/well of $^3$H-Thymidine was added and incubated for additional 16 h. Cell proliferation was indirectly quantified by measuring $^3$H-Thymidine incorporation. The stimulation index (SI) was obtained as the ratio between the mean counts *per* minutes (cpm) in each stimulated condition and the mean counts of the unstimulated cell.

**Transgene generation.** Codon-optimization of human BDD-FVIII transgene was performed by taking into consideration the codon-adaptation index, G/C content, Matrix Attachment Region-like sequences, destabilizing sequences, potential promoter binding sites, other *cis* acting negative regulatory elements, as described in the patent WO 2019/152692. The full sequences of coFVIII and coFVIII.XTEN are available in the patent WO 2019/152692 under the name of SEQ ID 71 and SEQ ID 72, respectively. XTEN 144 was inserted in the place of the B domain. The aminoacidic sequence of the XTEN polypeptide is:

GAPTSESATPESGPGSEPATSGSETPGTSESATPESGPGSEPATSGSETPGTSES ATPESGPGTSTEPSEGSAPGSPAGSPTSTEEGTSESATPESGPGSEPATSGS ETPGTSESATPESGPGSPAGSPTSTEEGSPAGSPTSTEEGASS. This design allows for the generation of a B-domain deleted activated FVIII without XTEN upon thrombin activation. The aminoacidic sequence of the FVIII-XTEN transgene is the following (the XTEN 144 polypeptide is underlined):

MQIELSTCFFLCLLRFCFSATRRYYLGAVELSWDYMQSDLGELPVDAR FPPRVPKSFPFNTSVVYKKTLFVEFTDHLFNIAKPRPPWMGLLGPTIQAEV YDTVVITLKNMASHPVSLHAVGVSYWKASEGAEYDDQTSQREKEDD KVFPGGSHTYVWQVLKENGPMASDPLCLTYSYLSHVDLVKDLNSGLIGA LLVCREGSLAKEKTQTLHKFILLFAVFDEGKSWHSETKNSLMQDRDAASARA WPKMHTVNGYVNRSLPGLIGCHRKSVYWHVIGMGTTPEVHSIFLEGHT FLVRNHRQASLEISPITFLTAQTLLMDLGQFLLFCHISSHQHDGMEAYVK VDSCPEEPQLRMKNNEEAEDYDDDLTDSEMDVVRFDDDNSPSFIQIRSVA KKHPKTWVHYIAAEEEDWDYAPLVLAPDDRSYKSQYLNNGPQRIGRKYK KVRFMAYTDETFKTREAIQHESGILGPLLYGEVGDTLLIIFKNQASRPYNIYPH GITDVRPLYSRRLPKGVKHLKDFPILPGEIFKYKWTVTVEDGPTKSDPRCLT RYYSSFVNMERDLASGLIGPLLICYKESVDQRGNQIMSDKRNVILFSVFDENR SWYLTENIQRFLPNPAGVQLEDPEFQASNIMHSINGYVFDSLQLSVCLHEVA YWYILSIGAQTDFLSVFFSGYTFKHKMVYEDTLTLFPFSGETVFMSMENPGLW ILGCHNSDFRNRGMTALLKVSSCDKNTGDYYEDSYEDISAYLLSKNNAI EPRSFSQNGAPTSESATPESGPGSEPATSGSETPGTSESATPESGPGSEPATSGS ETPGTSESATPESGPGTSTEPSEGSAPGSPAGSPTSTEEGTSESATPESGPGSEP ATSGSETPGTSESATPESGPGSPAGSPTSTEEGSPAGSPTSTEEGASSPPVLKRH QREITRTTLQSDQEEIDYDDTISVEMKKEDFDIYDEDENQSPRSFQKKTRHYF IAAVERLWDYGMSSSPHVLRNRAQSGSVPQFKKVVFQEFTDGSFTQPLYRGE LNEHLGLLGPYIRAEVEDNIMVTFRNQASRPYSFYSSLISYEEDQRQGAEPR KNFVKPNETKTYFWKVQHHMAPTKDEFDCKAWAYFSDVDLEKDVH SGLIGPLLVCHTNTLNPAHGRQVTVQEFALFFTIFDETKSWYFTENMERNC RAPCNIQMEDPTFKENYRFHAINGYIMDTLPGLVMAQDQRIRWYLLSMGSN ENIHSIHFSGHVFTVRKKEEYKMALYNLYPGVFETVEMLPSKAGIWRVECLIG EHLHAGMSTLFLVYSNKCQTPLGMASGHIRDFQITASGQYGQWAPKLARLH YSGSINAWSTKEPFSWIKVDLLAPMIIHGIKTQGARQKFSSLYISQFIIMYSLD GKKWQTYRGNSTGTLMVFFGNVDSSGIKHNIFNPPIIARYIRLHPTHYSI RSTLRMELMGCDLNSCSMPLGMESKAISDAQITASSYFTNMFATWSPSKAR LHLQGRSNAWRPQVNNPKEWLQVDFQKTMKVTGVTTQGVKSLLTSMYVK EFLISSSQDGHQWTLFFQNGKVKVFQGNQDSFTPVVNSLDPPLLTRYLRIHP QSWVHQIALRMEVLGCEAQDLY.

**Plasmid construction.** Plasmid pCCLsin.cPPT.ET.FVIII-BDD.142T, pCCLsin.cPPT.ET.coFVIII.142T and pCCLsin.cPPT.ET.coFIX.XTEN.142T were constructed by standard cloning techniques, by exchanging the gene synthesized different versions of the human FVIII cDNA (DNA 2.0) with FIX into the previously described pCCLsin.cPPT.ET.canineFIX.142T (NheI-SalI)[14].

**LV titration.** For LV titration, $1 \times 10^5$ 293T cells were transduced with serial LV dilutions in the presence of polybrene (8 μg/ml). Genomic DNA (gDNA) was extracted 14 days after transduction, using Maxwell 16 Cell DNA Purification Kit (Promega), following the manufacturer's instructions. VCN was determined by ddPCR, starting from 5–20 ng of template gDNA using primers (HIV fw: 5′-T ACTGACGGCTCTCGCACC-3′; HIV rv: 5′-TCTCGACGCAGGACTCG-3′) and a probe (FAM 5′-ATCTCTCTCCTTCTAGCCTC-3′) designed on the primer binding site region of LV. The amount of endogenous DNA was quantified by a primers/probe set designed on the human telomerase gene (Telo fw: 5′-GGCAC ACGTGGCTTTTCG-3′; Telo rv: 5′-GGTGAACCTCGTAAGTTTATGCAA-3′; Telo probe: VIC 5′-TCAGGACGTCGAGTGGACACGGTG-3′ TAMRA) or the human GAPDH gene (Applied Biosystems HS00483111_cm). The PCR reaction was performed with each primer (900 nM) and the probe (250 nM, 500 nM for Telo) following the manufacturer's instructions (Biorad), read with QX200 reader and analyzed with QuantaSoft software (Biorad). Infectious titer, expressed as TU/mL, was calculated using the formula TU/mL = (VCNx100,000x(1/dilution factor). LV physical particles were measured by HIV-1 Gag p24 antigen immunocapture assay (Perkin Elmer) following the manufacturer's instructions. LV-specific infectivity was calculated as the ratio between the infectious titer and physical particles.

**VCN determination.** For the experiment with Huh7 or Hepa1.6 cell lines, DNA was extracted using Maxwell 16 Tissue DNA Purification Kit (Promega), following the manufacturer's instructions. For mice experiments, DNA was extracted from whole liver or whole spleen samples using Maxwell® 16 Tissue DNA Purification Kit (Promega), DNA was extracted from fractionated/sorted liver cells using DNeasy Blood & Tissue Kit (Qiagen) or QIAamp DNA Micro Kit (Qiagen), according to cell number. VCN was determined in Huh7 samples as described above (see "LV titration"). VCN in murine DNA line was determined by ddPCR, starting from 5–20 ng of template gDNA using a primers/probe set designed on the primer binding site region of LV (see "LV titration" above). The amount of endogenous murine DNA was quantified by a primers/probe set designed on the murine sema3a gene (Sema3A fw: 5′-ACCGATTCCAGATGATTGGC-3′; Sema3A rv: 5′-TCCATATTAATGCAGTGCTTGC-3′; Sema3A probe: HEX 5′-AGAGGC CTGTCCTGCAGCTCATGG-3′ BHQ1). The PCR reaction was performed with each primer (900 nM) and the probe (250 nM) following the manufacturer's instructions (Biorad), read with QX200 reader and analyzed with QuantaSoft software (Biorad). VCN in NHP DNA was determined as described above (see "LV titration"). The amount of endogenous DNA was quantified by a primers/probe set designed on the TATA-Box Binding Protein Associated Factor 7 (*TAF7*) gene, (Applied Biosystems RH 02916247_s1), amplifying both the human and NHP *TAF7* genes. This analytical method was validated in compliance with the Organization for Economic Co-operation and Development (OECD) Principles of Good Laboratory Practice (GLP) in terms of accuracy (deviation vs. nominal VCN ≤ 15% until VCN = 0.007), specificity, intra-assay precision (coefficient of variation ≤ 3%), inter-assay precision (coefficient of variation ≤ 6%) and linearity ($R^2 ≥ 0.98$) within the range of 9.7 copies/reaction—28,776 copies/reaction for TAF7 and 4.8 copies/reaction—149,380 copies/reaction for HIV. The lower limit of VCN quantification was 0.007, which resulted within the accuracy criteria (deviation vs. nominal VCN ≤ 15%). The lower limit of VCN detection was 0.0003, calculated based on the lower limit of detection of the HIV system.

**Fractionation and sorting of liver cell subpopulations.** The mouse liver was perfused (15 mL/min) via the inferior vena cava with 12.5 mL of the following solutions at subsequent steps: (1) PBS EDTA (0.5 mM), (2) HBSS (Hank's balanced salt solution, Gibco) and HEPES (10 mM), 3) HBSS-HEPES 0.03% Collagenase IV (Sigma). The digested mouse liver tissue was harvested, passed through a 70 μm cell strainer (BD Biosciences) and processed into a single-cell suspension. This suspension was subsequently centrifuged three times (30, 25, and 20 *g*, for 3 min, at room temperature) to obtain hepatocytes-containing pellets. The nPC-containing supernatant was centrifuged (650 *g*, 7 min, at room temperature) and recovered cells were loaded onto a 30/60% Percoll (Sigma) gradient (1,800 *g*, for 20 min at room temperature). nPC interface was collected and washed twice. The nPC were subsequently incubated with the following monoclonal antibodies: e-fluor 450-conjugated anti-CD45 (30-F11, e-Bioscience), Allophycocyanin (APC)-conjugated anti-CD31 (MEC13.3, BD Biosciences), phycoerythrin (PE)-conjugated F4/80 (CI:A3-1, Biorad), PE-Cy5-conjugated anti-CD45R/B220 (from BD Biosciences), PE-Cy7-conjugated anti-CD11c (N418, e-Bioscience). nPC subpopulations (LSEC, KC, pDC) were sorted by FACS, BD FACSAria™ Fusion Cell Sorter (BD Biosciences); the nPC contaminating the PC suspension, were removed by FACS excluding cells labeled by APC-conjugated anti-CD31/anti-CD45 cocktail, thus obtaining sorted hepatocytes (Hep) using BD FACSMelody™ Cell Sorter (BD Biosciences). NHP liver pieces obtained at necropsy were chopped, washed twice in PBS and three times in washing medium (DMEM, ThermoFisher, 1% FBS,

Hyclone, penicillin and streptomycin 100 IU/mL, Lonza) and digested with 5 mL/g of digestion solution (EBSS, Hyclone, collagenase A 2.5 mg/mL, Sigma, dispase II 2.5 mg/mL, ThermoFisher) for 10 min at 37 °C. Then tissue was digested for an additional 2 h with 5 mL/g of digestion solution at 37 °C on a shaker (135-150 rpm). At the end of the digestion, cells were washed in cold washing medium (4 min at 120 g) and then the nPC-containing supernatant was processed as for the murine liver. After Percoll (Sigma) gradient separation, cells were washed twice and incubated with the following monoclonal antibodies: APC-conjugated anti-CD45 (D058-1283, BD pharmingen), FITC-conjugated anti-CD31 (WM59, BD pharmingen) and APC-H7-conjugated anti-HLA-DR (G46-6, BD pharmingen). nPC subpopulations (LSEC, KC) were sorted by FACS, BD FACSAria™ Fusion Cell Sorter (BD Biosciences). LSEC were sorted as CD45-negative/CD31-positive, while KCs were sorted as CD45-positive/HLA-DR-positive.

**Cell cultures and in vitro experiments.** 293T and Huh7 cell lines were maintained in Iscove's modified Dulbecco's medium (IMDM, Sigma) supplemented with 10% fetal bovine serum (FBS, Hyclone), 4 mM glutamine (Lonza), penicillin, and streptomycin 100 IU/mL (Lonza). Hepa1.6 cell line was maintained in Dulbecco's Modified Eagle Medium (DMEM, Sigma) supplemented with 10% fetal bovine serum (FBS, Hyclone), 4 mM glutamine (Lonza), penicillin and streptomycin 100 IU/mL (Lonza). All cells were maintained in a 5% $CO_2$ humidified atmosphere at 37 °C. All cell lines were routinely tested for mycoplasma contamination. Huh7 or Hepa1.6 cell lines were transduced for 24 h then cultured for 14 days before gDNA extraction and VCN determination (see "VCN determination"). 250,000 cells were then plated and after 72 h the supernatant was collected and hFVIII concentration was assessed by ELISA (see "FVIII assays"). Cell lines were originally obtained from ATCC.

**Flow cytometry.** Flow cytometry analyses were performed using a FACSCanto analyzer (BD Biosciences), equipped with DIVA Software. Blood was collected from mice from retro-orbital plexus and 20 μL were directly stained with 20 μL of antibody mix (see below for details) for 20 min at 4 °C in the dark. Cells were then fixed with 500 μL of Lyse/Fix Buffer 1X (BD Biosciences, Phosflow 558049) for 10 min at 37 °C, washed with PBS or MACS buffer (PBS pH 7.2 0.5% BSA, 2 mM EDTA), treated with Fc Receptor-Block (Miltenyi Biotec) and then re-suspended in the buffer used for washing.

Anti-mouse CD3 BUV737, clone 17A2 (eBioscience, 1:250)
Anti-mouse CD45 PE-Cy7, clone 30-F11 (Invitrogen, 1:250)
Anti-mouse CD4 BV786, clone RM4-4 (BD Biosciences, 1:250)
Anti-mouse CD8 PB, clone 53-6.7 (BD Biosciences, 1:250)
Anti-mouse CD19 APC, clone RA3-6B2 (BD Biosciences, 1:250)
Anti-mouse B220 PE, clone 1D3 (BD Biosciences, 1:250)
Anti-human CD4 FITC, clone MT477 (BD Biosciences, 1:100)
Anti-human CD8 SK1 PerCP-Cy5.5, clone SK1(Biolegend, 1:100)

**Cytokine and anti-VSV.G Abs ELISA.** The concentrations of cytokines and chemokines were determined in NHP serum by a magnetic-based multiplex ELISA 26 analytes (NHP XL Cytokine Luminex Performance Premixed Panel, R&D systems or Human cytokine A Premixed Magnetic Luminex Performance Assay, R&D systems, for IL18 and IL1-RA), following the manufacturer's instructions. The concentrations of anti-VSV.G Abs were determined in NHP serum by ELISA, coating with recombinant VSV.G (Alpha Diagnostic Intl.) 1 μg/mL and developing with a HRP conjugated mouse anti-monkey IgG Ab (Southern Biotech 1:10,000).

**RNA extraction and ddPCR.** RNA extraction was performed using Maxwell 16 LEV simplyRNA Tissue Kit (Promega), according to the manufacturer's instructions and reverse transcribed using the SuperScript IV VILO kit (11766050; ThermoFisher Scientific). LV gene expression was assessed by ddPCR starting from 25 to 50 ng of template cDNA using a primers/probe set designed on the WPRE region of LV (WPRE: primer fw 5′-GGCTGTTGGGCACTGACAAT-3′; primer rv 5′-ACGTCCCGCGCAGAATC-3′; probe FAM 5′-TTTCCTTGGCTGC TCGCCTGTGT-3′ NGB). Taf7 was used as reference gene (see "VCN determination") The PCR reaction was performed with each primer (900 nM) and the probe (250 nM) following the manufacturer's instructions (Biorad), read with QX200 reader and analyzed with QuantaSoft software (Biorad).

**Analysis of half-life of FVIII protein.** 293 cells were transfected with plasmid encoding B-domain deleted FVIII with or without XTEN using PEI transfection (ThermoFisher Scientific) following the manufacturer's instruction. At 72 h post transfection, cell culture supernatants were harvested and concentrated 30-fold using Millipore Amicon Ultra-15 centrifuge filter unit following 0.2 μm filtration. Concentrated culture mediums were then administered i.v. to hemophilia A mice, plasma samples were then collected at indicated time points for FVIII activity assay.

**Statistical analysis.** Statistical analyses were performed upon consulting with professional statisticians at the San Raffaele University Center for Statistics in the Biomedical Sciences (CUSSB). When normality assumptions were not met, non-parametric statistical tests were performed. Two-tailed Mann–Whitney test was performed to compare two independent groups, while in presence of more than two independent groups Kruskal–Wallis test followed by post-hoc analysis (Dunn's test for multiple comparisons against the reference control group along with Bonferroni's correction) was applied. The strength of the relationship between two quantitative variables was assessed by Spearman's rank-order test. Expression of FVIII in different treatment groups and over time (starting from the steady-state) was modeled using two-tailed Linear Mixed Effects (LME) models. Group/dosage indicator variable and time variable were included in the model, both as main effects as well as in interaction. This modeling approach allows to properly capture the dependency structure among observations arising from measuring multiple times the expression levels in the same mouse. To account for mouse-specific heterogeneity, a random effect was specified on mouse ID, hence random intercept models were estimated. Cubic and square root transformations of the outcome were also considered to satisfy underlying model assumptions. After model estimation, post-hoc analyses have been implemented to compare experimental treatment groups to the reference control group at a fixed time point (Table S12). LMEs were estimated in R (version 4.0.3) by means of the *nlme* package (Pinheiro J, Bates D, DebRoy S, Sarkar D, R Core Team (2020). nlme: Linear and Nonlinear Mixed Effects Models. R package version 3.1-148, https://CRAN.R-project.org/package=nlme; Russell V. Lenth (2021). emmeans: Estimated Marginal Means, aka Least-Squares Means. R package version 1.5.4, https://CRAN.R-project.org/package=emmeans).

## Data availability

The LV and reagents described in this manuscript are available to interested scientists upon signing a MTA with standard provisions. There are some restrictions on the use of the provided materials in research involving LV-based gene therapy of hemophilia, except for research aimed at reproducing the findings reported in this manuscript, according to the collaboration agreement between Fondazione Telethon, San Raffaele Scientific Institute and Bioverativ/Sanofi. All data associated with this study are available in the main text or the in the Supplementary Information/Source data file. Source data are provided with this paper.

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

## Acknowledgements

This work was supported by Telethon (SR-Tiget Core Grant 2011-2016) and Bioverativ/Sanofi sponsored research agreement. We thank CUSSB for statistical consulting, MolMed S.p.A. for large-scale production and purification of the LV batches used in the NHP study, and all members of the Cantore and Naldini laboratory for helpful discussions. C.C. conducted this study as partial fulfillment of his International Ph.D. Course in Molecular Medicine at San Raffaele University, Milan.

## Author contributions

M.M. designed and performed experiments, analyzed and interpreted data and wrote the manuscript. C.C. and T.L. designed and performed experiments, analyzed data and edited the manuscript. M.B., F.R., T.P., R.C., S. P-W., D.D., I.V. performed experiments and analyzed data. C.B. performed statistical analysis. P.A. supervised I.V. work. A.F. provided crucial reagents and intellectual input. E.A. coordinated experiments with NHP. C.M. supervised T.L. research. A.A. performed experiments, analyzed and interpreted data related to immune responses and edited the manuscript. L.N. supervised research, interpreted data and edited the manuscript. A.C. supervised and coordinated research, interpreted data and wrote the manuscript.

## Competing interests

L.N., A.C., A.A., M.M., T.L., S.P.W. are inventors on patent applications submitted by Foundation Telethon and San Raffaele Scientific Institute or Bioverativ/Sanofi on LV technology related to the work presented in this manuscript (WO2019/152692; WO2016009326). FT and SRSI, through SR-Tiget, have established a research collaboration on liver-directed lentiviral gene therapy of hemophilia with Bioverativ/Sanofi. The remaining authors declare no competing interests.
