## [Peer Review File · Nature Communications]

Reviewers' Comments:

Reviewer #1:

Remarks to the Author:

The manuscript by Milani and coworkers continues a long standing investigation of in vivo intravenous lentiviral gene therapy for hemophilia A. The authors demonstrated both coFVII and coFVIII.XTEN constructs offered large increases in protein production in vitro and in vivo, including a therapeutic response.

The introduction is not balanced. While pointing out issues of AAV gene therapy, they point out only the advantages of lentivirus, not the disadvantages. More balance is needed in this preclinical manuscript. Also, the question of transgenic FVIII duration should be briefly covered, although the reviewer recognizes published data are scarce. The 5 year AAV5-FVIII results were published this year (Pasi et al) and give the best published example of the progressive loss of FVIII activity although Spark data presented at meetings also show loss of FVIII in some patients using AAV-LKO3-FVIII. The relationship of exogenous FVIII expression in a heterologous cell has not been ruled out as a contributor to the loss, which may be of relevance to this work.

In the long term experiment, a lower dose was used than used in the neonatal experiment, and a sub therapeutic amount of FVIII (5-10%) was found at the end of 1.5 years. Despite the fact that the authors say this was stable, It is hard to find a pattern to the different doses used and some explanation should be more clear.

The XTEN peptide, currently being explored clinically at the protein level, is not well described. How many different constructs were tried. Why was the B-domain selected for insertion? How does this compare to the work done on protein constructs? Understanding whether the large increase is due to increased transcription or increased half life is important and has implications for the variability that is seen. Understanding the relative contributions, and extrapolating from prior studies by some of the authors at the protein level would be most informative. The precise nucleotide break for BDD and XTEN insertion sites should be presented in S1. XTEN represents different sequences and different lengths. Please be specific about what was tested.

Fig 1 suggests there is no advantage of XTENylating coFVIII, at least in these neonatal experiments. The tail clip assay is notoriously variable, can be better controlled by more precise amputation (noted a range of amputations in mm was performed), and really requires a larger N to make any quantitative statements. The authors should comment on the lack of differences between the two experimental groups. The tail clip assay may have been more discriminatory than an amputation model.

Fig 2, the diminution of activity but not antigen (a and b) is perplexing, ie 80 weeks and 20 weeks are the same for antigen, but 80 weeks is half the 20 week activity. Likewise, the loss of hepatocyte expression (Fig 2c vs 1e [note different doses] vs Fig 2h at 6 months suggests loss of VCN. There does not appear to be a dose response curve looking at 2h. I presume Fig 2abc are all from the same mice, harvested at 20 months? Uniform labeling (eg, weeks) should be used. Removing 4 outliers in 2e defeats the purpose of understanding the variability, a key point as the authors contemplate clinical translation, especially when such small numbers within each group are presented. They should go back in or at least be represented in the figure legend.

A and b from Fig 2 and 3 would benefit from showing the individual data points. The authors use SE, and the results indicate the SD are quite high. Are the individual mice consistent, or variable over time? Why was such a high dose given to the RagHemoA mice compared to the earlier experiments? For the R593C mice, assume the high dose was given to push an antibody response?

Although I find Fig4h most interesting, I'm not sure it is appropriate to select peak values (assume at different times and plot them). I don't know what peak values mean, but they are not of long term therapeutic benefit, thus a dose response in steady state is more relevant for clinical translation. If the goal is to look at FVIII levels prior to antibody generation, this is not clear from the figure, and it's hard to read the Nab levels- the graph really should go from 0-10, with a cut mark for the two higher ones as you have done. In general some graphs are difficult to read and

would benefit from an expanded y-axis at lower values, which are of most interest.

Fig 5c highlights the question of where the coFVIII went? Some explanation is needed since I'm not clear XTEN should have any effect on integrated VCN? The N values in these experiments are a concern, which are N=3 or 2. It is very difficult to draw any conclusions when sample sizes are this small and inherent variability is so high.

The hypothesis or hypotheses behind the cytokine panel is not well described. A high viral load is entering the body in an unconventional way- what should be expected in the first day or 2? Is this consistent with an immune response to contain the virus? I'm not sure Fig 6 adds much to the discussion and may be more appropriate for supplemental.

S7 really points to the heterogeneity seen in these experiments. The authors don't discuss the causes, possible solutions, yet shouldn't this be considered prior to human experimentation. A better focus and sense for the variability is needed in this manuscript.

The authors have assembled a tremendous amount of data, and have generated new concepts (ie, XTEN, FVIII specific mutation mice). The abstract reflects none of this. The authors should concentrate less on AAV in the abstract and more on the original data they generated, which advances the field.

Minor points:

P3, line 63- AAV5 has 20-30% seroprevalence since it is a goat AAV (Arbetman et al J Virol 2005). The other natural AAVs derived from humans and NHP are closer to 50-60% seroprevalence. In this section, AAV is liver toxic, etiology not elucidated for FVIII and other diseases such as X-lined myotubular myopathy. Not addressed by authors.

P4, lines 75-81 are confusing re: ECs. While the statements are true, they imply the authors are targeting sinusoidal EC, so clarification should be made.

P14, lines 332-335 is not a clear sentence and should be rewritten.

P15, lines 359-361: Many would no longer call 12% the "minimal desirable target". Minimal, perhaps, but not desirable since joint damage will continue to occur at that level in individuals with hemophilic arthropathy. Ref 17 is incorrect here.

In the discussion on insertional mutagenesis, some discussion of the Bluebird results would be appropriate along with a more balanced discussion.

P21 line 493, more description of hFVIII-XTEN should be provided since this is a novel construct.

P21 line 514-518: the cutoff for the linear portion for the Bethesda assay should be provided. Assume it is 1 BU, which is not particularly sensitive for low titer antibodies but in any event should be specified.

P22 line 533, why were two different steroid regimens used?

In Fig 4ab, an the authors comment on the AST spike in the absence of an ALT spike? Could it be from muscle due to handling during the procedure? Any muscle specific enzymes (eg CPK) elevated that would confirm muscle as the source (Supplemental Tables 2-11 show CPK elevations)?

Fig 5b, why the large variability between different pieces of the liver. Suggests at least 10x differences within a single animal for mRNA expression. I note the reference range for ALT, 16-120, is the highest I have seen. Please double check the upper limit.

Fig S2b, I probably missed this, but some detail on the purified hepatocytes or at least an indication where this is more fully described in the text. These are difficult experiments to do given the rapidly declining health of the hepatocytes. It may be worth considering putting this figure into the main body.

Fig S4a- while I see the correlation between AST and p24, I'm not convinced this is due to liver. Is LV going into skeletal muscle? Have you looked? In the rest of S4, what is causing Hgb and HCT to go down? Blood loss would be my guess. From blood draws or due to Nabs? Fig S4g might be more informative on a semi-log plot, as has been done for AAV clearance.

Reviewer #2:

Remarks to the Author:

In this study, Milani et al. report the use of alloantigen-free and phagocytosis-shielded lentivirus vectors (LV) to deliver engineered versions of human FVIII transgenes to hemophilia A mice and nonhuman primates (NHPs). The authors show that a codon-optimized version of FVIII containing an unstructured XTEN polypeptide resulted in significantly greater levels of transgene expression than FVIII transgenes lacking codon-optimization and the XTEN peptide in murine and human cell lines. Similar results were observed in hemophilia A mice, where improved therapeutic efficacy by the co.FVIII.XTEN transgene was also demonstrated. The murine studies also showed the negative impact of anti-drug antibodies on transgene expression, as improved levels of FVIII were detected in immunodeficient or tolerized animals. The authors then tested the LV.coFVIII and LV.coFVIII.XTEN vectors in NHPs, which had to be put on an immunosuppressive regimen to allow detection of human FVIII in the absence of ADAs. Here again, levels of coFVIII.XTEN surpassed those of the coFVIII, although expression of these proteins went down after the immunosuppressive regimen was lifted and ADAs emerged. Up until this point, the conclusions from this paper seemed logical and were corroborated by the data presented. But then the authors turned to speculation to explain some of the unexpected findings of their immunological analyses.

Overall, this is an interesting study and the improved levels and persistence of hFVIII expression achieved with the LV.coFVIII.XTEN vector in NHPs are noteworthy. However, excessive speculations about the results and the substandard reporting of the T-cell data in NHPs constitute the main weakness of this paper. Some specific points are below.

1) the inverse relationship between peak cytokine concentrations and LV particle dose was attributed to the "accumulation of CD47 signaling on professional phagocytes" suppressing cytokine release in animals receiving the highest doses of the LV.coFVIII vectors; there is no evidence in the paper to suggest such a mechanism.

2) The IFN-gamma ELISPOST data in PBMC (Suppl. Fig. 6b) do not support the claim that monkeys treated with the FVIII.XTEN transgene showed lower hFVIII-specific T-cell responses than monkeys treated with FVIII. Although Figure 6K may support this conclusion in splenocytes, those responses range between 2.0 (two) and 7.0 (seven) spot-forming cells (SFC)/10⁶ splenocytes. This is way below the typical lower limit of detection (50 SFC/10⁶ cells) for IFN-gamma ELISPOT assays. Besides, no positive controls are shown for the ELISPOT assays, so there is no way to judge the dynamic range of the assay. The lack of positive controls is particularly aggravating for panels c and d of Suppl. Fig. 6, which show absence of detectable IFN-gamma and IL-5 production in hepatic LN cells and splenocytes from LV.coFVIII-treated animals. How do we know that those assays even worked?

3) The exaggerated speculation goes on in following sentences: "Conversely, IFN-gamma-producing T cells were detected in response to FVIII stimulation in splenocytes of NHP treated with LV.coFVIII (3/5), while only a weak response was present in 1/5 NHP treated with LV.coFVIII.XTEN (Fig. 6k). This rapid in vitro response upon FVIII peptide stimulation (24 hours of stimulation) correlates with the lower residual transgene expression in the liver (see Fig. 5b), therefore likely derived from memory FVIII-specific cytotoxic T cells that mediated elimination of FVIII-expressing hepatocytes". So is the idea here is that hFVIII-specific T-cells in the spleen are reducing transgene expression in the liver? Note that hFVIII-specific T-cells were not detected in the hepatic LNs (Suppl. Fig. 6c) and these responses were equivalent in PBMC from animals in the two groups. So why would FVIII-specific cytotoxic T-cells eliminate FVIII-expressing hepatocytes in LV.coFVIII-treated animals but not in the LV.coFVIII.XTEN-treated group?

4) Fig. 5b: Why do authors attribute the increased T-cell proliferation detected in this assay as coming from CD4+ T-helper cells? Why not CD8+ T-cells? In any event, the proliferative data from the positive control in this assay must be displayed to rule out the possibility that the difference in cell viability detected between the two groups is due to greater cell viability in the LV.coFVIII-treated animals.

5) The higher levels of FVIII transgene RNA at necropsy in the LV FVIII.XTEN group is indeed intriguing. However, the claim that the XTEN polypeptide might have reduced the immunogenicity of the FVIII seems too speculative, considering that XTEN-specific immune responses were not

assessed in the study.

Reviewer #3:

Remarks to the Author:

Milani and colleagues have previously demonstrated phenotypic correction of hemophilia B in mice and dogs following successful lentivirus (LV) mediated efficient and long-term gene transfer to the liver. In continuation of this work, they now describe in this manuscript their lentiviral-mediated liver directed gene therapy strategy to achieve correction of hemophilia A in mice and non-human primates (NHP). Using the allo-antigen free and phagocytosis-shielded (CD47hi) LV they developed recently, they have established delivery of a codon optimized B-domain deleted (BDD)-Factor VIII (FVIII) transgene containing an XTEN polypeptide (LV.coFVIII-XTEN) that resulted in long term FVIII activity in blood and phenotypic correction of hemophilia A mice. They also show therapeutic-range human FVIII activity in NHP following immunosuppression. A novel double KO mouse model combining the hemophilia and immune deficient phenotypes that could be useful in pre-clinical investigation of gene transfer and editing strategies was also generated during the course of this study.

This is an impressive study with a systematic approach that is well executed, carefully interpreted and lucidly presented.

There remain a few issues that need clarification:

I think the authors have not fully highlighted the promise of a LV strategy as it pertains to a truly curative gene therapy with pediatric application. The limitation of AAV-mediated liver-directed gene therapy is not simply the waning FVIII expression in hemophilia A adults and the difficult challenge of readministration. Rather, gene therapy as applied only in adults (or even adolescents) is unlikely to change the phenotype of this bleeding disorder with respect to the development of joint disease later in life – note the disappointing results from the Joint Outcome Study-Continuation recently reported where progressive joint disease is observed even with prophylaxis applied from early childhood. An integrating strategy as applied through LV-mediated liver transduction offers real hope to permanently alter the phenotype of this disorder by preventing bleeding as early in life as possible.

From Fig.1C, it appears that the FVIII-Ab development sets in much earlier with LV-coFVIII-XTEN (~4 weeks post injection) when compared with either LV-FVIII or LV-coFVIII (starting at ~12 weeks post injection) and seems much more pronounced as well. This contrasts with the observations in the NHP model. Do the authors have an explanation for this apparent difference in the immune response with inclusion of the XTEN sequence?

Fig.1D does not show any data points pertaining to LV.FVIII-Abs. Were there no survivors at this time point?

It is interesting to note that LV.FVIII treated mice fared even more poorly than Hemophilia A mice in the hemostatic challenge assay (Fig.1D) – although I don't see that statistical calculation was applied. Even with only about 20% of normal FVIII expression, these LV.FVIII treated mice are expected to do much better than Hem A mice. Was this a consistent observation across multiple experiments?

In their description of data in Fig.2, the authors indicate that ~25 weeks was the longest analyzed and FVIII output seemed stable except for those that developed anti-FVIII Abs. But the data in Fig.2A,B shows stable LV.coFVIII expression even at 80 weeks post LV treatment. Does this mean that LV.coFVIII-XTEN was more immunogenic than LV.coFVIII and that the anti-FVIII antibody development was much quicker just as was seen in Fig.1C?

Fig.3 includes data only for LV.coFVIII-XTEN in the adult hemophilia immunodeficient RagHemoA and immune competent HemoA-R593C. Was a comparative analysis with LV.co-FVIII not necessary?

Also, the rationale for using half the dosage (4×10^{10} TU/kg vs. 8×10^{10} TU/kg) for the HemoA-R593C is not explained. Considering that the expression was still stable at 25 weeks post LV, an

extended analysis until and beyond 45 weeks, as in the case with RagHemoA mice, might have been useful.

The standard error for the data in Fig.3 D, E seem unusually large.

The authors argue that inclusion of the XTEN polypeptide in the FVIII transgene might have alleviated some of its immunogenicity in the NHP model tested in this study. However, the authors have not provided any insights as to what mechanism the XTEN confers its improved plasma levels. The XTEN polypeptide may have uncoupled FVIII from the VWF chaperone effect that otherwise constrains its pharmacokinetics in circulation. However, it is also well established that VWF plays a crucial role in FVIII immunogenicity and protects FVIII from inhibitor inactivation. Given these two seemingly contradictory scenarios, the authors should provide a more robust evidence-backed explanation for their observations.

The authors have indicated that the allo-antigen free and phagocytosis-shielded (CD47hi) LV developed and used by them in the current study has shown no evidence of acute toxicity or clonal expansion of transduced cells. A recent report by Nguyen et al (2021, Nature Biotechnology) on a 10 year follow-up study of AAV gene therapy in dogs with hemophilia A has identified clonal expansion of transduced liver cells and therefore advises caution and stresses the importance of long term monitoring for potential genotoxicity. Do the authors have any reassuring data from similar long term follow up of their earlier studies on LV gene therapy in dogs and NHPs with hemophilia B?

The results in NHP show the challenging immune response to the human FVIII used in these studies. However, other investigators have suggested a tolerizing effect that can be seen with continued expression even after early appearance of neutralizing antibodies. Do the authors have any non-immunosuppressed NHPs followed over a longer period of time to test whether there is any evidence for tolerization (understanding the challenge of likely cross-reactivity with endogenous FVIII in the NHPs).

As acknowledged in the introduction, most current gene therapy strategies target hepatocytes to produce transgenic FVIII even though endothelial cells have been identified as the natural source of FVIII. The authors acknowledge that stability and turnover post-natally and in adulthood are not known. Nevertheless, what drawbacks or significant challenges do the authors foresee in adapting their current LV strategy to safely and effectively target hepatic endothelial cells for FVIII transgene expression?

Reviewer #1:

The manuscript by Milani and coworkers continues a long standing investigation of in vivo intravenous lentiviral gene therapy for hemophilia A. The authors demonstrated both coFVII and coFVIII.XTEN constructs offered large increases in protein production in vitro and in vivo, including a therapeutic response.

The introduction is not balanced. While pointing out issues of AAV gene therapy, they point out only the advantages of lentivirus, not the disadvantages. More balance is needed in this preclinical manuscript.

We agree with the Reviewer that the disadvantages of lentiviral vectors (LV) were not highlighted in the introduction, however, please note that these were mentioned in the discussion. To follow the Reviewer suggestion, we have now included the following sentence in the introduction of the revised manuscript: “Absence of prior clinical testing of systemic administration, manufacturing hurdles and concerns about insertional mutagenesis have until now hindered pre-clinical development of *in vivo* LV gene therapy directed to the liver”.

Also, the question of transgenic FVIII duration should be briefly covered, although the reviewer recognizes published data are scarce. The 5 year AAV5-FVIII results were published this year (Pasi et al) and give the best published example of the progressive loss of FVIII activity although Spark data presented at meetings also show loss of FVIII in some patients using AAV-LKO3-FVIII. The relationship of exogenous FVIII expression in a heterologous cell has not been ruled out as a contributor to the loss, which may be of relevance to this work.

We agree with the Reviewer that the reasons underlying the decrease of transgenic FVIII in AAV vector gene therapy are neither fully understood nor extensively reported. We have expanded the sentence in which we mention this outcome in the revised manuscript, which now reads: “However, a decreasing trend in FVIII transgene expression has been reported, for reasons that are not fully understood and might be related to the challenge of stably maintaining functional episomal vector genomes reaching up to their packaging limit”.

In the long term experiment, a lower dose was used than used in the neonatal experiment, and a sub therapeutic amount of FVIII (5-10%) was found at the end of 1.5 years. Despite the fact that the authors say this was stable, It is hard to find a pattern to the different doses used and some explanation should be more clear.

We thank the Reviewer for this comment. We have tried to clarify the rationale for dose selection in the revised manuscript. Moreover, we performed an additional LV-dose response experiment for the codon-optimized FVIII transgene without XTEN (new Fig. 2e-g and new Fig. 2j), in order to clarify both the relationship between the administered LV dose and FVIII transgene output and to better compare the two FVIII transgenes at multiple LV doses. This new LV-dose response experiment confirms our previous data, by showing that the LV-dose response regarding FVIII output is not linear for the codon-optimized FVIII transgene and that the difference between this LV and that carrying the XTENylated FVIII transgene is more evident at lower LV doses, providing normal-range FVIII amounts.

The XTEN peptide, currently being explored clinically at the protein level, is not well described. How many different constructs were tried. Why was the B-domain selected for insertion? How does this compare to the work done on protein constructs? Understanding

whether the large increase is due to increased transcription or increased half life is important and has implications for the variability that is seen. Understanding the relative contributions, and extrapolating from prior studies by some of the authors at the protein level would be most informative. The precise nucleotide break for BDD and XTEN insertion sites should be presented in S1. XTEN represents different sequences and different lengths. Please be specific about what was tested.

We would like to point out that all the sequences of the transgenes used in this work are disclosed in the patent WO 2019/152692, as mentioned in the supplementary methods. However, we now also include the amino acid sequence of the XTEN 144 polypeptide and of the entire FVIII.XTEN transgene in the supplementary methods. XTEN 144 was inserted in place of the B domain, similarly to the protein product BIVV001 (Konkle *et al.*, *N Engl J Med* 2020), which however also contains additional domains. This design allows for the generation of a B-domain deleted activated FVIII without XTEN, upon thrombin activation. Regarding mechanisms of improved FVIII transgene output by XTENylation, it is known that inclusion of the XTEN polypeptide increases the half-life of the modified protein, thus resulting in higher steady-state amounts of the transgene product. Improved secretion may also contribute to the observed outcome. Transcription is unlikely to be involved, since all the LV tested in this work share the promoter/enhancer sequences. We now also include data about half-life of FVIII protein with or without XTEN, confirming increased half-life by the XTENylated FVIII which remained 10-fold higher 2 days after administration to hemophilia A mice. These data are shown in new Supplementary Fig. 2d of the revised manuscript.

Fig 1 suggests there is no advantage of XTENylating coFVIII, at least in these neonatal experiments. The tail clip assay is notoriously variable, can be better controlled by more precise amputation (noted a range of amputations in mm was performed), and really requires a larger N to make any quantitative statements. The authors should comment on the lack of differences between the two experimental groups. The tail clip assay may have been more discriminatory than an amputation model.

We agree with the Reviewer that the experiment shown in Fig. 1 does not show major differences between coFVIII and coFVIII.XTEN, however a 2-fold higher circulating FVIII amount by the XTEN-containing transgene becomes apparent if normalized on LV copies in sorted hepatocytes (Supplementary Fig. 2a). To further investigate the difference between the two transgenes we performed a new LV-dose response experiment with LV expressing the coFVIII transgene (new Fig. 2e-g) and compared the results with the previously performed LV-dose response experiment carried out with the coFVIII.XTEN expressing LV (new Fig. 2j). These new data show that the coFVIII.XTEN steady-state amounts were approximately between 2- and 7-fold higher than coFVIII at similar LV doses and that a smaller difference in FVIII amounts between the 2 transgenes was observed at the highest LV doses, in line with the previous experiments. Regarding the hemostasis challenge, we agree with the Reviewer that the tail clip assay is poorly quantitative. Please note that we performed this assay to assess restoration of functional coagulation and to highlight the difference between the wild-type and codon-optimized FVIII transgenes. Indeed, all the mice treated with either coFVIII- or coFVIII.XTEN expressing LV that achieved supranormal FVIII activity (Fig. 1a and b) regained fully normal hemostasis, while hemostasis remained defective in mice treated with LV.FVIII that only expressed approximately 20 % of normal FVIII on average.

Fig 2, the diminution of activity but not antigen (a and b) is perplexing, ie 80 weeks and 20 weeks are the same for antigen, but 80 weeks is half the 20 week activity. Likewise, the loss

of hepatocyte expression (Fig 2c vs 1e [note different doses] vs Fig 2h at 6 months suggests loss of VCN. There does not appear to be a dose response curve looking at 2h. I presume Fig 2abc are all from the same mice, harvested at 20 months? Uniform labeling (eg, weeks) should be used.

We thank the Reviewer for this comment. Please note that there is some mouse-to-mouse variability in FVIII transgene expression and activity. To better represent the outcome of this experiment, and as suggested by the Reviewer, we now show single mouse data in new Fig. 2 a, b and c instead of mean and variance. The new data presentation highlights that only one mouse clearly displays decreasing FVIII at later time points. We also include new data about anti-FVIII antibodies and show only low-level anti-FVIII antibodies. We also modified the description of these results in the revised manuscript, which now reads: “We showed FVIII blood concentration and activity that remained stable throughout the nearly lifetime follow-up at around 5-10% of normal on average (Fig. 2a, b). All these mice had low amounts of anti-FVIII Abs (Fig. 2c), with 3/4 maintaining stable FVIII activity and 1/4 showing a decreasing trend at the latest time points.” Regarding differences in VCN, please note that former Fig. 2c now Fig 2d showed VCN in purified hepatocytes, while VCN in former Fig. 2h now Fig 2l was determined in total liver (thus comprising VCN of different cell types), thus cannot be directly compared. To better clarify this aspect, we have now changed the labelling of the y axis of Fig 2l, which now reads “Total liver VCN”.

Removing 4 outliers in 2e defeats the purpose of understanding the variability, a key point as the authors contemplate clinical translation, especially when such small numbers within each group are presented. They should go back in or at least be represented in the figure legend.

Please note that no data was excluded from the graphs. The mice that developed anti-FVIII antibodies and those that did not were displayed separately in former Fig. 2g now Fig 2k to better visualize the dose-response data in former Fig. 2d, e now Fig. 2h, i.

A and b from Fig 2 and 3 would benefit from showing the individual data points. The authors use SE, and the results indicate the SD are quite high. Are the individual mice consistent, or variable over time? Why was such a high dose given to the RagHemoA mice compared to the earlier experiments? For the R593C mice, assume the high dose was given to push an antibody response?

We thank you the Reviewer for this comment. We now show individual mouse data in new Fig. 2 and 3 of the revised manuscript. Regarding the high dose used in the adult mice, please note that newborn mice are more permissive to transduction, thus need lower LV doses than adults to achieve similar transgene output.

Although I find Fig4h most interesting, I’m not sure it is appropriate to select peak values (assume at different times and plot them). I don’t know what peak values mean, but they are not of long term therapeutic benefit, thus a dose response in steady state is more relevant for clinical translation. If the goal is to look at FVIII levels prior to antibody generation, this is not clear from the figure, and it’s hard to read the Nab levels- the graph really should go from 0-10, with a cut mark for the two higher ones as you have done. In general some graphs are difficult to read and would benefit from an expanded y-axis at lower values, which are of most interest.

We followed the Reviewer suggestion and now show average of FVIII activity before the development of antibodies (NHP for NHP) in new Fig. 4h of the revised manuscript. We now also split the y axis of Fig. 4 to better highlight values in the lower range.

Fig 5c highlights the question of where the coFVIII went? Some explanation is needed since I'm not clear XTEN should have any effect on integrated VCN? The N values in these experiments are a concern, which are N=3 or 2. It is very difficult to draw any conclusions when sample sizes are this small and inherent variability is so high.

Please note that Fig. 5c shows VCN in different liver cell types and highlights low VCN in Kupffer cells, due to the reduction of LV phagocytosis mediated by high surface content of CD47. Please also note that VCN at end-point is expected also to reflect possible cytotoxic immune responses against transduced cells expressing FVIII-derived antigens (i.e. hepatocytes), thus supporting our contention of lower immune response to XTEN-carrying FVIII transgene, given the higher end-point VCN in hepatocytes in NHP treated with LV.coFVIII-XTEN vs. LV.coFVIII.

The hypothesis or hypotheses behind the cytokine panel is not well described. A high viral load is entering the body in an unconventional way- what should be expected in the first day or 2? Is this consistent with an immune response to contain the virus? I'm not sure Fig 6 adds much to the discussion and may be more appropriate for supplemental.

We agree with the Reviewer that an inflammatory response to systemic administration of a high load of viral particles is expected. However, please note that we analyzed a wide panel of different cytokines to better understand the type of acute innate immune reactions to LV administration and possible correlations with later-occurring adaptive immune responses. Since cytokine responses are also considered a safety-related read-out, we prefer to keep these data within the main figures.

S7 really points to the heterogeneity seen in these experiments. The authors don't discuss the causes, possible solutions, yet shouldn't this be considered prior to human experimentation. A better focus and sense for the variability is needed in this manuscript.

Please note that Supplementary Fig. 7 is designed to correlate data of innate and adaptive immune responses, FVIII expression at peak and endpoint. Indeed, we would like to point out that most of the variability in the NHP experiment is likely related to the development of immune responses against the human FVIII, impacting on remaining transgene expression at the end of the follow up. Higher variability in FVIII output in adult mice compared to newborn mice may be due to lower permissiveness to transduction of hepatocytes in the former. We have included this last sentence in the results section of the revised manuscript.

The authors have assembled a tremendous amount of data, and have generated new concepts (ie, XTEN, FVIII specific mutation mice). The abstract reflects none of this. The authors should concentrate less on AAV in the abstract and more on the original data they generated, which advances the field.

We thank the Reviewer for the positive comments. We have now modified the abstract to better reflect the content of our manuscript.

Minor points:

P3, line 63- AAV5 has 20-30% seroprevalence since it is a goat AAV (Arbetman et al J Virol 2005). The other natural AAVs derived from humans and NHP are closer to 50-60% seroprevalence. In this section, AAV is liver toxic, etiology not elucidated for FVIII and other diseases such as X-linked myotubular myopathy. Not addressed by authors.

We agree with the Reviewer that AAV gene therapy has been correlated with some liver toxicity, however, as the Reviewer points out, the aetiology is not fully understood yet. We prefer not to discuss AAV gene therapy in details, since it may be considered out-of-scope for this manuscript.

P4, lines 75-81 are confusing re: ECs. While the statements are true, they imply the authors are targeting sinusoidal EC, so clarification should be made.

We agree with the Reviewer and have now modified this part of the introduction in the revised manuscript which now reads: “It has been shown that the natural source of most FVIII production is endothelial cells. Gene transfer of LV expressing FVIII from liver endothelial cells has been proposed and some encouraging results have been reported in hemophilia A mice treated as adults, however the stability and turnover of these cells, both in post-natal liver growth and homeostasis in adulthood remain not fully understood. Indeed, currently the most clinically advanced AAV-based gene therapy strategies and the LV-based strategy described in this work exploit hepatocytes to produce transgenic FVIII”.

P14, lines 332-335 is not a clear sentence and should be rewritten.

We thank the Reviewer for this comment and have now changed this sentence in the discussion of the revised manuscript, which now reads: “Long-term stability of FVIII-XTEN transgene expression and activity in both mice treated as newborns and adults indicates absence of selective disadvantage of transduced hepatocytes, which may be due to toxicity from FVIII overexpression. On the same line, absence of increase in transgene over time indicates lack of expansion of transduced hepatocytes, which may potentially be triggered by LV insertional mutagenesis”.

P15, lines 359-361: Many would no longer call 12% the “minimal desirable target”. Minimal, perhaps, but not desirable since joint damage will continue to occur at that level in individuals with hemophilic arthropathy. Ref 17 is incorrect here.

We thank the Reviewer for this comment. We have now changed this sentence in the discussion of the revised manuscript, which now reads: “Restoration of FVIII activity above 12% of normal in people with severe hemophilia A is considered adequate to mostly prevent joint hemorrhages and may thus be set as the minimal target for correction in gene therapy”. We have also corrected the reference.

In the discussion on insertional mutagenesis, some discussion of the Bluebird results would be appropriate along with a more balanced discussion.

We agree with the Reviewer and have now added a sentence in the discussion of the revised manuscript, which reads: “clonal expansions leading in 2 patients to myelodysplastic syndrome were recently reported in a single trial using LV with strong viral enhancer/promoter”.

P21 line 493, more description of hFVIII-XTEN should be provided since this is a novel construct.

We agree with the Reviewer and now provide more description of the FVIII-XTEN construct both in main and supplementary materials and methods sections.

P21 line 514-518: the cutoff for the linear portion for the Bethesda assay should be provided. Assume it is 1 BU, which is not particularly sensitive for low titer antibodies but in any event should be specified.

We consider as cutoff value 1 BU/mL, calculated on mean + 2 standard deviations on 10 pre-gene therapy samples (shown in Fig. 4j, -14 days). We have now added this information in the materials and methods section of the revised manuscript.

P22 line 533, why were two different steroid regimens used?

In the second and third group of treated NHP (lower LV doses) the window of immune suppression was slightly enlarged, to avoid impaired detection of the transgenic human FVIII due to low-level anti-human FVIII antibodies.

In Fig 4ab, an the authors comment on the AST spike in the absence of an ALT spike? Could it be from muscle due to handling during the procedure? Any muscle specific enzymes (eg CPK) elevated that would confirm muscle as the source (Supplemental Tables 2-11 show CPK elevations)?

We agree with the Reviewer and have now added a sentence in the results of the revised manuscript, which reads: “AST elevation positively correlated with the dose of LV particles infused (Supplementary Fig. 4a), however it may also be related to muscle damage during animal handling for sample collection, since serum creatin phosphokinases were also elevated (Supplementary Tables 2-11).”

Fig 5b, why the large variability between different pieces of the liver. Suggests at least 10x differences within a single animal for mRNA expression.

Please note that maximum difference within a single animal is 2-fold in Fig. 5b.

I note the reference range for ALT, 16-120, is the highest I have seen. Please double check the upper limit.

Please note that the reference range for ALT is that available for *Macaca Fascicularis*.

Fig S2b, I probably missed this, but some detail on the purified hepatocytes or at least an indication where this is more fully described in the text. These are difficult experiments to do given the rapidly declining health of the hepatocytes. It may be worth considering putting this figure into the main body.

Please note that an updated description of the fractionation and sorting of liver cell sub-populations is given in the supplementary materials and methods, in addition to what we have previously described in Milani *et al.*, Sci Transl Med 2019.

Fig S4a- while I see the correlation between AST and p24, I'm not convinced this is due to liver. Is LV going into skeletal muscle? Have you looked? In the rest of S4, what is causing Hgb and HCT to go down? Blood loss would be my guess. From blood draws or due to Nabs?

Regarding AST elevation, please see our response to the point above. We did not find LV VCN in the muscle both in this (Fig. 5a) and our previous work (Milani *et al.*, *Sci Transl Med* 2019). Blood loss due to anti-NHP FVIII cross-reacting antibodies is the likely explanation for the decrease in Hgb and HCT, as mentioned in the results section of the manuscript.

Fig S4g might be more informative on a semi-log plot, as has been done for AAV clearance.

We prefer to keep the linear scale for the y axis, since the log scale does not allow plotting the 0 value.

Reviewer #2:

In this study, Milani *et al.* report the use of alloantigen-free and phagocytosis-shielded lentivirus vectors (LV) to deliver engineered versions of human FVIII transgenes to hemophilia A mice and nonhuman primates (NHPs). The authors show that a codon-optimized version of FVIII containing an unstructured XTEN polypeptide resulted in significantly greater levels of transgene expression than FVIII transgenes lacking codon-optimization and the XTEN peptide in murine and human cell lines. Similar results were observed in hemophilia A mice, where improved therapeutic efficacy by the co.FVIII.XTEN transgene was also demonstrated. The murine studies also showed the negative impact of anti-drug antibodies on transgene expression, as improved levels of FVIII were detected in immunodeficient or tolerized animals. The authors then tested the LV.coFVIII and LV.coFVIII.XTEN vectors in NHPs, which had to be put on an immunosuppressive regimen to allow detection of human FVIII in the absence of ADAs. Here again, levels of coFVIII.XTEN surpassed those of the coFVIII, although expression of these proteins went down after the immunosuppressive regimen was lifted and ADAs emerged. Up until this point, the conclusions from this paper seemed logical and were corroborated by the data presented. But then then the authors turned to speculation to explain some of the unexpected findings of their immunological analyses.

Overall, this is an interesting study and the improved levels and persistence of hFVIII expression achieved with the LV.coFVIII.XTEN vector in NHPs are noteworthy. However, excessive speculations about the results and the substandard reporting of the T-cell data in NHPs constitute the main weakness of this paper. Some specific points are below.

1) the inverse relationship between peak cytokine concentrations and LV particle dose was attributed to the “accumulation of CD47 signaling on professional phagocytes” suppressing cytokine release in animals receiving the highest doses of the LV.coFVIII vectors; there is no evidence in the paper to suggest such a mechanism.

We agree with the Reviewer that there is no direct experimental evidence to support this mechanism in this work. Please note, however, that in our previously published work (Milani *et al.*, *Sci Transl Med* 2019), we reported a “paradox” effect in which LV with high CD47 surface content (CD47^{hi}-LV) was more protected from phagocytosis at lower doses compared to higher doses, suggesting that increasing CD47 signaling on phagocytes at high LV particle

load resulted in overall decrease of phagocytosis even of LV with basal CD47 content. We suggest that a similar mechanism may result in reduced innate immune response by increasing signaling mediated by CD47. Please also note that these data are reported in a Supplementary Figure and since we consider of interest this unexpected observation, we prefer to provide a possible explanation, even if speculative.

2) The IFN-gamma ELISPOST data in PBMC (Suppl. Fig. 6b) do not support the claim that monkeys treated with the FVIII.XTEN transgene showed lower hFVIII-specific T-cell responses than monkeys treated with FVIII. Although Figure 6K may support this conclusion in splenocytes, those responses range between 2.0 (two) and 7.0 (seven) spot-forming cells (SFC)/ 10^6 splenocytes. This is way below the typical lower limit of detection (50 SFC/ 10^6 cells) for IFN-gamma ELISPOT assays. Besides, no positive controls are shown for the ELISPOT assays, so there is no way to judge the dynamic range of the assay. The lack positive controls is particularly aggravating for panels c and d of Suppl. Fig. 6, which show absence of detectable IFN-gamma and IL-5 production in hepatic LN cells and splenocytes from LV.coFVIII-treated animals. How do we know that those assays even worked?

We thank the Reviewer for careful revision of immunological analyses included in our study and to underline several points that need further explanation and omissions in the presentation of the data that are now resolved.

To clarify our data interpretation and support the strength of the presented data several details need to be considered.

Studies conducted in patients and primates treated with AAV gene therapy postulated a threshold positivity (50 SFC/ 10^6 cell) and the extent of a positive response based on the in vitro antigenic stimulation with peptide pools covering the sequence of the relevant AAV capsid. However, several important details may explain differences in the frequencies of FVIII reactive cells observed in our studies:

1. To our knowledge this is the first report of quantification of IFN γ releasing cells in response to FVIII derived epitopes in the spleen and lymph-nodes of NHP treated with LV gene therapy, therefore neither the extent of the response (if any) nor the background noise of IFN γ release were known.
2. We studied a response evoked by the expression of the LV encoded FVIII in hepatocytes of normal non-hemophilic NHPs. This represented a completely different scenario compared to an immune response to epitopes derived from AAV particles, provided systemically at high doses and cannot be compared to any conventional pathogen encounter.
3. Our threshold for positivity, although stringent, is conventionally accepted in biological assays. We calculate the average number of SFC obtained in multiple negative control wells, exposing cell to only Dimethyl sulfoxide (DMSO, solvent for peptide) without any antigen, increased by 2 standard deviations. Thus, we assigned a threshold for each tested cell sample, which may significantly differ as shown in panel B of the figure below.
4. Overall, the estimated frequency of IFN γ releasing T cells in response to FVIII (by IFN γ ELISpot assay) ranges between 1 and 26 SFU/ 10^6 splenic T cells, which is not so unconventional. These values have been calculated according to percentages of T cells in the spleen determined by FACS analysis (~50%, see new Supplementary Fig. 7). Therefore, the frequency of FVIII specific memory T cells (2.6×10^5 and 1×10^6) appears to be realistic in comparison to those typically achieved in other conditions (see additional figure for Reviewer).

Besides these comments, we followed the Reviewer suggestions by improving the reporting of ELISpot results and adding an entire Supplementary Figure. As the Reviewer can appreciate in the figure below and in new Supplementary Fig. 7, we now report representative images of IFN γ and IL-5 ELISpot assays and positive controls of T-cell stimulation in ELISpot assay. We always included a polyclonal stimulation in the assay as positive control (TPA/ionomycin) to confirm cell viability and potential capacity of releasing the cytokines of interest upon stimulation, and a negative control (DMSO) to set the threshold of positivity for each single NHP of the study, as previously mentioned. Overall, we are confident that data reported in Fig. 6k were properly generated using ELISpot as the gold standard assay for the identification of rare antigen-specific T cells. Therefore, we consider the observed frequency of FVIII responsive cells, although very low, representative of the memory FVIII-specific T cell response.

Figure for Reviewer 2:

Cellular response to transgene in secondary lymphoid organs. Cells from secondary lymphoid organs from LV.FVIII treated NHPs were harvested at the end of experiments and stored alive in liquid nitrogen. At thawing, splenic and hepatic lymph-node (LN) cells were stimulated in culture by overlapping peptides (15aa long, 5aa off-set) covering the entire sequence of FVIII.XTEN (1 μ M each) to determine reactivity to the transgene. Peptide reported in red belongs to XTEN region (A). Alternatively, TPA/ionomycin or DMSO replicates were added as positive control (T-cell polyclonal stimulation) and negative control (peptide diluent), respectively. Representative IFN γ ELISpot assays of splenocytes (B), hepatic LN (C) and IL-5 ELISpot assays of splenocytes (D) are reported.

3) The exaggerated speculation goes on in following sentences: “Conversely, IFN-gamma-producing T cells were detected in response to FVIII stimulation in splenocytes of NHP treated with LV.coFVIII (3/5), while only a weak response was present in 1/5 NHP treated with LV.coFVIII.XTEN (Fig. 6k). This rapid in vitro response upon FVIII peptide stimulation (24 hours of stimulation) correlates with the lower residual transgene expression in the liver (see Fig. 5b), therefore likely derived from memory FVIII-specific cytotoxic T cells that mediated elimination of FVIII-expressing hepatocytes”. So is the idea here is that hFVIII-specific T-cells in the spleen are reducing transgene expression in the liver? Note that hFVIII-specific T-cells were not detected in the hepatic LNs (Suppl. Fig. 6c) and these responses were equivalent in PBMC from animals in the two groups. So why would FVIII-specific cytotoxic T-cells eliminate FVIII-expressing hepatocytes in LV.coFVIII-treated animals but not in the LV.coFVIII.XTEN-treated group?

We understand the criticisms regarding these speculations and acknowledge the challenge of evaluating immune responses in large animal models. However, we are confident that clarifications provided above convinced the Reviewer that our interpretation of the data is supported and not exaggerated.

We discussed results also considering our prior experience accumulated in the analysis of anti-transgene immune response as correlated with the success or failure of an LV based *in vivo* gene therapy in multiple preclinical models. Therefore, we considered that the correlation between persistence of coFVIII.XTEN transcription and reduced response to transgene in LV.coFVIII.XTEN-treated NHP was relevant and should be highlighted to the readers. This observation appears to be in line with our previous results. Indeed, our previous studies showed that a reduced transgene-specific T cell response in the spleen is often associated with establishment of long-term expression of the transgene in hepatocytes and possibly of tolerance. Moreover, we demonstrated that transgene-responsive cells may permanently co-exist with the transgene expression in hepatocytes depending on several factors (transgene immunogenicity, percentage of transgene-expressing hepatocytes, generation of regulatory response, tolerance induction; Annoni *et al.* Blood 2009, Matrai *et al.* Hepatology 2011, Annoni *et al.* EMBO Mol Med 2013).

A possible explanation for the absence of the response in hepatic lymph-nodes may derive from a very low frequency of these cells (below the limit of ELISpot detection), while being barely detectable in the spleen.

While extension of protein half-life by XTENylation is proven and understood, little is known about reduction of immunogenicity of XTENylated antigens (Schellenberger *et al.* Nat Biotechnol 2009, Podust *et al.* J of Controlled Release 2016). We can speculate that an extended half-life in the blood of the protein might correlate with a reduction of its uptake and presentation to T cells. Moreover, the establishment of a constant elevated concentration of an antigen *in vivo* may result in tolerization rather than effective immune response.

In addition, this is the first, to our knowledge, in-depth analysis of anti-transgene immune response induced by transgenic expression of two variants of FVIII expressed in hepatocytes of normal non-hemophilic NHP. Specifically, little is known on T-cell memory formation and homing to lymphoid organs occurring after antigenic priming in NHP and/or humans treated with a hepatocyte-directed gene therapy.

Of note, there is consensus that 24-hour stimulation is sufficient to determine the release of IFN γ by antigen-specific memory CD8 T cells stimulated *in vitro* with the cognate antigen and, at the same time minimizing background signal.

Therefore, we consider that our speculations based on immunological analyses and transgene expression data are of interest and relevant to identify advantages of expressing XTENylated transgenic factors for the correction of coagulation disorders by *in vivo* gene therapy.

4) Fig. 5b: Why do authors attribute the increased T-cell proliferation detected in this assay as coming from CD4+ T-helper cells? Why not CD8+ T-cells? In any event, the proliferative data from the positive control in this assay must be displayed to rule out the possibility that the difference in cell viability detected between the two groups is due to greater cell viability in the LV.coFVIII-treated animals.

T-cell proliferation was measured stimulating total splenocytes at the indicated doses of FVIII protein. Proliferative response is driven by antigen recognition presented by splenic antigen presenting cells. In this experimental setting the vast majority of antigen presentation occurs in the context of class-II Major Histocompatibility Complex (MHC-II), since the antigen is taken up from extracellular medium. Events of antigen cross-presentation in the context of MHC-I to CD8 T cell are not absent, but rare in comparison. However, we recognize that in the absence of a formal prove we have to ascribe the proliferation to the entire T cell population, as we now refer to in the revised manuscript.

We included (NHP specific, Mabtech clone CD3-1) anti-CD3 polyclonal stimulation. Unfortunately, in terms of cpm, this kind of positive control is not informative, but useful in case of studies related to cytokines release. Indeed, kinetics of proliferation driven by anti-CD3 and FVIII antigenic stimulation were extremely different, when 3H-thymidine was added (day 5) anti-CD3 stimulated T cells were mostly dead by the constant and excessive stimulation, as expected. However, viability of splenic T cells is demonstrated by massive release of IFN γ upon TPA/ionomycin stimulation of total splenocytes, as previously mentioned and now shown in Supplementary Fig. 7.

5) The higher levels of FVIII transgene RNA at necropsy in the LV FVIII.XTEN group is indeed intriguing. However, the claim that the XTEN polypeptide might have reduced the immunogenicity of the FVIII seems too speculative, considering that XTEN-specific immune responses were not assessed in the study.

We thank the Reviewer for recognizing the relevance of our observation that at necropsy, far

from the immunosuppressive regimen, levels of FVIII transgene RNA were higher in the LV.coFVIII.XTEN-treated than LV.coFVIII-treated groups. However, in support to our data interpretation and discussion about a potential role of the XTEN polypeptide in prolongation of transgene expression by reducing the immune response to transgenic FVIII expressing hepatocytes, there are several direct and indirect evidences to be considered.

XTENylation of the transgene is the only variation between the two groups of treatment. Therefore, it is conceivable that XTEN insertion played a role in the prolongation of transgene expression.

In support of this interpretation, Shellenberg *et al.* (Nat Biotechnol 2009) and the new data included in the revised manuscript (new Supplementary Fig. 2d) showed increased half-life of the protein in the circulation, which might correlate with a reduction of its uptake and presentation to T cells. Moreover, the establishment of a constant elevated concentration of an antigen *in vivo* may result in tolerization rather than effective immune response. Furthermore, the absence within XTEN polypeptide of hydrophobic amino acids, which typically contribute to compact structures and/or protein aggregation, minimizes the possibility to generate new T cell epitopes. Indeed, we did not identify a significant response by antigenic stimulation with peptide pools 9, 10 and 11, which comprise the XTEN region.

Reviewer #3 (Remarks to the Author):

Milani and colleagues have previously demonstrated phenotypic correction of hemophilia B in mice and dogs following successful lentivirus (LV) mediated efficient and long-term gene transfer to the liver. In continuation of this work, they now describe in this manuscript their lentiviral-mediated liver directed gene therapy strategy to achieve correction of hemophilia A in mice and non-human primates (NHP). Using the allo-antigen free and phagocytosis-shielded (CD47hi) LV they developed recently, they have established delivery of a codon optimized B-domain deleted (BDD)-Factor VIII (FVIII) transgene containing an XTEN polypeptide (LV.coFVIII-XTEN) that resulted in long term FVIII activity in blood and phenotypic correction of hemophilia A mice. They also show therapeutic-range human FVIII activity in NHP following immunosuppression. A novel double KO mouse model combining the hemophilia and immune deficient phenotypes that could be useful in pre-clinical investigation of gene transfer and editing strategies was also generated during the course of this study. This is an impressive study with a systematic approach that is well executed, carefully interpreted and lucidly presented.

There remain a few issues that need clarification:

I think the authors have not fully highlighted the promise of a LV strategy as it pertains to a truly curative gene therapy with pediatric application. The limitation of AAV-mediated liver-directed gene therapy is not simply the waning FVIII expression in hemophilia A adults and the difficult challenge of readministration. Rather, gene therapy as applied only in adults (or even adolescents) is unlikely to change the phenotype of this bleeding disorder with respect to the development of joint disease later in life – note the disappointing results from the Joint Outcome Study-Continuation recently reported where progressive joint disease is observed even with prophylaxis applied from early childhood. An integrating strategy as applied through LV-mediated liver transduction offers real hope to permanently alter the phenotype of this disorder by preventing bleeding as early in life as possible.

We thank the Reviewer for the positive feedback. We followed the Reviewer suggestion and now included a sentence about this important point in the discussion of the revised manuscript that reads: “In humans, restoration of stable, therapeutic amounts of FVIII activity since early childhood may be preferred to avoid accumulation of joint damage, by preventing bleeding as early in life as possible”.

From Fig.1C, it appears that the FVIII-Ab development sets in much earlier with LV-coFVIII-XTEN (~4 weeks post injection) when compared with either LV-FVIII or LV-coFVIII (starting at ~12 weeks post injection) and seems much more pronounced as well. This contrasts with the observations in the NHP model. Do the authors have an explanation for this apparent difference in the immune response with inclusion of the XTEN sequence?

Please note that mice in Fig. 1 were treated as newborns and in the X axis we always show the weeks post LV administration. Anti-FVIII Abs start to appear late in both groups, during adulthood, at 12 weeks post LV in the LV.coFVIII.XTEN-treated group (only 1 mouse out of 9) and at 20 weeks post LV in the LV.coFVIII-treated group. Despite the development of low level anti-FVIII Abs, all the mice maintained FVIII activity as shown by hemostatic challenge (Fig. 1d). For these reasons, we would not consider FVIII.XTEN as more immunogenic in this setting.

Fig.1D does not show any data points pertaining to LV.FVIII-Abs. Were there no survivors at this time point? It is interesting to note that LV.FVIII treated mice fared even more poorly than Hemophilia A mice in the hemostatic challenge assay (Fig.1D) – although I don't see that statistical calculation was applied. Even with only about 20% of normal FVIII expression, these LV.FVIII treated mice are expected to do much better than Hem A mice. Was this a consistent observation across multiple experiments?

We would like to point out that there are no points in the LV.FVIII-Abs group because no mice in this group developed anti-FVIII Abs. Regarding the multiple comparisons performed in the statistical analysis, we compared all the datasets to wt controls, in order to statistically test if any group was significantly different from the wt. We chose to apply this type of analysis because with these relatively low number of replicates we do not have the proper power to compare all datasets together. Finally, we would like to mention that this specific tail clipping assay is very stringent and only mice with high FVIII levels display restoration of hemostasis. There are other assays, such as the tail vein transection assay, in which lower amounts of FVIII are enough to observe bleeding differences compared to hemophilia A mice.

In their description of data in Fig.2, the authors indicate that ~25 weeks was the longest analyzed and FVIII output seemed stable except for those that developed anti-FVIII Abs. But the data in Fig.2A,B shows stable LV.coFVIII expression even at 80 weeks post LV treatment. Does this mean that LV.coFVIII-XTEN was more immunogenic than LV.coFVIII and that the anti-FVIII antibody development was much quicker just as was seen in Fig.1C?

We thank the Reviewer for this comment. The longest time-point analyzed in the experiment shown in former Fig. 2d-h, now Fig. 2h-l, was indeed 25 weeks post LV administration, thus we do not have longer follow-up for this LV in mice treated as newborns. However, please note that we show stable FVIII output in adult mice treated with LV.coFVIII.XTEN until 43 weeks after LV administration in HemoA-Rag immunodeficient mice (Fig. 3a-c) and now also in HemoA-R593C immunocompetent mice, for which we now included two additional time points at 32 and 42 weeks after LV administration (Fig. 3d-f). Regarding the immunogenicity

of LV.coFVIII, we now included anti-FVIII Abs measurement in mice shown in Fig. 2a, b and show in the new Fig. 2c of the revised manuscript that indeed also in these mice we can measure low-level Abs, that only in 1 mouse out of 4 impacted on the stability of FVIII antigen and activity. Please note that, in these mice, Abs start to be measurable 4 weeks post LV administration. Taken together these data do not show major differences in terms of immunogenicity of LV.coFVIII and LV.coFVIII.XTEN in hemophilia A mice, but highlights the increased FVIII output mediated by XTENylation, in the range of doses that allow reconstitution of physiological amounts of FVIII activity, as we now report in the new Fig. 2j.

Fig.3 includes data only for LV.coFVIII-XTEN in the adult hemophilia immunodeficient RagHemoA and immune competent HemoA-R593C. Was a comparative analysis with LV.co-FVIII not necessary?

As mentioned in the response above, the higher amounts of circulating FVIII obtained in mice treated with LV.coFVIII.XTEN prompted us to test this construct in the adult setting. Please note that higher doses are needed when treating adult mice compared to newborns to achieve similar amounts of circulating FVIII, thus we treated adult mice only with the best performing LV construct.

Also, the rationale for using half the dosage (4×10^{10} TU/kg vs. 8×10^{10} TU/kg) for the HemoA-R593C is not explained. Considering that the expression was still stable at 25 weeks post LV, an extended analysis until and beyond 45 weeks, as in the case with RagHemoA mice, might have been useful.

We thank the Reviewer for this comment. As mentioned above, we now included other two time points in the follow-up of HemoA-R593C mice (33 and 42 weeks post LV, Fig. 3d-e). that were still alive at the time of the first submission. The LV dose was reduced, since supraphysiologic FVIII amounts were observed in some mice treated at the higher LV dose.

The standard error for the data in Fig.3 D, E seem unusually large. The authors argue that inclusion of the XTEN polypeptide in the FVIII transgene might have alleviated some of its immunogenicity in the NHP model tested in this study. However, the authors have not provided any insights as to what mechanism the XTEN confers its improved plasma levels. The XTEN polypeptide may have uncoupled FVIII from the VWF chaperone effect that otherwise constrains its pharmacokinetics in circulation. However, it is also well established that VWF plays a crucial role in FVIII immunogenicity and protects FVIII from inhibitor inactivation. Given these two seemingly contradictory scenarios, the authors should provide a more robust evidence-backed explanation for their observations.

We thank the Reviewer for this comment. We now plot single-animal values in Fig. 3 for both experiments. XTENylation of proteins has been previously reported to increase half-life of the payload protein. We now also included a new experiment in which we administered intravenously either FVIII-BDD protein or FVIII.XTEN protein in hemophilia A mice, measured FVIII activity overtime and showed prolonged half-life of the latter (new Supplementary Fig. 2d), suggesting that progressive accumulation of the transgenic FVIII-XTEN results in higher steady-state FVIII output. Regarding the second point, even if we do not have a direct proof of maintained interaction with the VWF we should take into consideration that VWF is known to bind the C1 domain, with the C2, A3 domains and the a3 acidic peptide containing additional sites implicated in the FVIII-VWF interaction (Chiu *et al.* Blood 2015) and all these domains are maintained in our FVIII.XTEN construct, since the

XTEN 144 polypeptide was inserted in place of the B domain, similarly to the protein product BIVV001 (Konkle *et al.*, N Engl J Med 2020), which however also contains additional domains. We now also include the amino acid sequence of the XTEN 144 polypeptide and of the entire FVIII.XTEN transgene in the supplementary methods.

The authors have indicated that the allo-antigen free and phagocytosis-shielded (CD47hi) LV developed and used by them in the current study has shown no evidence of acute toxicity or clonal expansion of transduced cells. A recent report by Nguyen *et al.* (2021, Nature Biotechnology) on a 10 year follow-up study of AAV gene therapy in dogs with hemophilia A has identified clonal expansion of transduced liver cells and therefore advises caution and stresses the importance of long term monitoring for potential genotoxicity. Do the authors have any reassuring data from similar long term follow up of their earlier studies on LV gene therapy in dogs and NHPs with hemophilia B?

We thank the Reviewer for this comment. Please note that we recently published a paper in Nature Medicine (Cesana *et al.*, Nat Med 2021) where we report LV integration site (IS) analysis performed on cell-free DNA released into the bloodstream and on liver DNA of LV-treated hemophilia B dogs (Cantore *et al.*, Sci Transl Med 2015). In this work we show a polyclonal repertoire of genetically modified cells without dominant clones expanding and persisting over time. We now also reference this work in the discussion of the revised manuscript.

The results in NHP show the challenging immune response to the human FVIII used in these studies. However, other investigators have suggested a tolerizing effect that can be seen with continued expression even after early appearance of neutralizing antibodies. Do the authors have any non-immunosuppressed NHPs followed over a longer period of time to test whether there is any evidence for tolerization (understanding the challenge of likely cross-reactivity with endogenous FVIII in the NHPs).

Unfortunately, we do not have any longer-term follow-up in the NHP and we are aware that other pre-clinical studies are needed. In particular, as we mention in the manuscript the anti-FVIII Abs developed were cross-reacting with the NHP FVIII and the phenotype reported in Supplementary Fig. 4b-e is consistent with acquired hemophilia A that for ethical reasons precluded long-term follow up of these NHP.

As acknowledged in the introduction, most current gene therapy strategies target hepatocytes to produce transgenic FVIII even though endothelial cells have been identified as the natural source of FVIII. The authors acknowledge that stability and turnover post-natally and in adulthood are not known. Nevertheless, what drawbacks or significant challenges do the authors foresee in adapting their current LV strategy to safely and effectively target hepatic endothelial cells for FVIII transgene expression?

At the moment, hepatocyte gene transfer to produce FVIII is the most used strategy in gene therapy for hemophilia A, both in pre-clinical models and in clinical trials. As mentioned in the introduction of the manuscript “Gene transfer of LV expressing FVIII from liver endothelial cells has been proposed and some encouraging results have been reported in hemophilia A mice treated as adults (Merlin *et al.*, Mol Ther 2017; Merlin *et al.*, Blood Adv 2019)”, thus LV are a suitable vehicle to transfer genes to endothelial cells and the encoded expression cassette can be engineered to specifically target expression to these cells. However, no data are available about stability after gene transfer to newborns to evaluate maintenance of

therapeutic FVIII levels following growth. Moreover, no data in large animals are available up to now, thus efficacy in NHP should be evaluated and long-term data about stability should be generated before translating this strategy to humans.

Reviewers' Comments:

Reviewer #1:

Remarks to the Author:

The authors have responded satisfactorily to the issues I raised.

Reviewer #2:

Remarks to the Author:

The authors provided a detailed reply to my original comments, especially those concerning the analysis of T-cell responses performed in the study. While I acknowledge the difficulties in quantifying tissue-specific T-cell responses in NHPs, especially when working with cryopreserved cells, I am still not concerned with the stringency of the T-cell analysis. Most of the animals analyzed in suppl fig 6b show IFN-gamma+ T-cell responses that were too low in frequency and detected at only one or two time points.

The way the data are presented is problematic because the heat maps do not reveal the background levels of IFN-gamma production in the no stim wells, which was substantial in hepatocytes from some of the animals, as shown in the figure included in the rebuttal.

In their rebuttal, the authors claim that "the estimated frequency of IFNg releasing T cells in response to FVIII (by IFNg ELISpot assay) ranges between 1 and 26 SFU/10⁶ splenic T cells, which is not so unconventional." Not so unconventional? I worry about anyone claiming that they can look at an ELISPOT well and tell the difference between the background well and one containing "one" or "two" FVIII-specific T-cell. Consider the ELISPOT picture included in the rebuttal file. For two of the three animals, the background is almost indistinguishable from the treatment wells.

I disagree with the authors that IFN-gamma-producing T-cell ranging in frequency from 1 (one!!!) to 26 cells out of a million can be accurately detected by IFN-g ELISPOT using *cryopreserved samples*. As shown in the paper by Reynolds et al. (PMID: 22569124), cryopreservation results in major reductions in both the magnitude and frequency of T-cell responses measured by ELISPOT, which is particularly problematic for detecting low frequency T-cell responses.

The T-cell measurements in this paper remain problematic.

Reviewer #3:

Remarks to the Author:

The authors have satisfactorily addressed all the points raised in the previous critique.

Reviewer #2:

The authors provided a detailed reply to my original comments, especially those concerning the analysis of T-cell responses performed in the study. While I acknowledge the difficulties in quantifying tissue-specific T-cell responses in NHPs, especially when working with cryopreserved cells, I am still not concerned with the stringency of the T-cell analysis. Most of the animals analyzed in suppl fig 6b show IFN-gamma+ T-cell responses that were too low in frequency and detected at only one or two time points.

We agree with the Reviewer that the T-cell responses observed in the PBMC and reported in Supplementary Figure 6b were of low frequency. However, please note that we only consider positive those samples 2 standard deviations above average background wells, as stated in the materials and methods of the manuscript and in the response to referee letter, thus allowing sufficient stringency to consider those responses real, based on the assay used to detect them. Please note that treated animals have their own endogenous FVIII, thus we cannot expect immune responses of high intensity, such as those evoked by viruses or virus-derived vectors, e.g. AAV vectors. Indeed, as shown in our point-by-point response, the estimated frequencies of these antigen-specific T cell responses are in the upper range of those observed in the case of immune responses to autoantigens, thus in line with our experimental setting. Even if of low frequency, we remain confident that the data regarding anti-FVIII transgene immune responses in the context of systemic *in vivo* LV gene therapy are novel and of interest for the field. We acknowledge the Reviewer's point that T-cell responses in the PBMC were detected at few time points and have now rephrases the description of these results in the text of the revised manuscript as follows: "We observed a delayed transient anti-FVIII T cell response in PBMC which became detectable from day 21 post LV, mainly in NHP expressing the highest FVIII amounts, with the overall tendency to contract at undetectable level at the end of the follow up, 60 days post LV (in 3/5 of LV.coFVIII-treated NHP and 4/5 of LV.coFVIII.XTEN-treated NHP; Supplementary Fig. 6b)" changed to: "We observed low-frequency sporadic anti-FVIII T cell responses in PBMC in all treatment groups (Supplementary Fig. 6b)".

The way the data are presented is problematic because the heat maps do not reveal the background levels of IFN-gamma production in the no stim wells, which was substantial in hepatocytes from some of the animals, as shown in the figure included in the rebuttal.

Please note that all the data tables with the actual numbers underlying the heat maps are available in the source data files. We now also include: i) a data table with the spleen ELISPOT raw data even before background subtraction in the source data file; ii) all the pictures of the spleen ELISPOT wells in revised Supplementary Figure 7c.

In their rebuttal, the authors claim that "the estimated frequency of IFN γ releasing T cells in response to FVIII (by IFN γ ELISpot assay) ranges between 1 and 26 SFU/10⁶ splenic T cells, which is not so unconventional." Not so unconventional? I worry about anyone claiming that they can look at an ELISPOT well and tell the difference between the background well and one containing "one" or "two" FVIII-specific T-cell. Consider the ELISPOT picture included in the rebuttal file. For two of the three animals, the background is almost indistinguishable from the treatment wells. I disagree with the authors that IFN-gamma-producing T-cell ranging in frequency from 1 (one!!!) to 26 cells out of a million can be accurately detected by IFN-g ELISPOT using

cryopreserved samples. As shown in the paper by Reynolds et al. (PMID: 22569124), cryopreservation results in major reductions in both the magnitude and frequency of T-cell responses measured by ELISPOT, which is particularly problematic for detecting low frequency T-cell responses. The T-cell measurements in this paper remain problematic.

Please note that the indicated 1-26 spot forming units (SFU)/10e6 T cells is the general range of the data shown in Fig. 6k related to T-cell responses in the spleen at end-point and derives from a mathematical calculation. We concur with the Reviewer that we could not distinguish one spot above background in the assay, we only consider positive those samples 2 standard deviations above average of 6 background wells. The data that we consider reliable and on which we base our conclusion are well above 10 SFU/10e6 T cells. There is just one data point with 1 SFU/10e6 T cells, that is actually almost invisible in the heat map, and indeed it falls into the FVIII-XTEN group, for which we suggest reduced immunogenicity compared to FVIII without XTEN. Moreover, please consider that the actual anti-FVIII cellular response in each NHP corresponds to the sum of the number of T cells responsive in all peptide pools covering the entire FVIII sequence (see figure below). Please note that the data indeed look quite clear to us, as the only positive signals are found in T cells of NHP treated with the LV encoding the FVIII transgene without XTEN.

Please consider that presentation of the T-cell responses in the manuscript is meant to be mostly descriptive and suggesting that the FVIII-XTEN transgene may be slightly less immunogenic compared to the FVIII without XTEN. This conclusion is also based on remaining FVIII transgene in the circulation and vector RNA in the liver at the end of the follow up, besides T-cell responses in the spleen. Nevertheless, we have been more cautious and added in the discussion of the revised manuscript a statement as follows: “The overall low frequency of anti-FVIII T cell responses detected in this study suggests that more sensitive immune assays are needed to confirm these data and further assess anti-transgene immune responses in future studies carried out in similar experimental settings”. Overall, we are confident that the T-cell responses observed in our study are real, even if low, well detectable by the assay used and in line with the expected frequency reported in literature for immune responses to moderate antigens such as autoantigens, which is similar to

expressing a human FVIII in NHP having their own monkey FVIII. We consider the way we analyze these data is stringent and appropriate and that our conclusion is supported by multiple data of the NHP study besides T-cell responses. We also deem that our assessment of anti-transgene immune response is of interest and informative for the field.